



# Parameter uncertainty dominates C cycle forecast errors over most of Brazil for the 21st Century

Thomas Luke Smallman[1,2], David Thomas Milodowski[1,2], Eráclito Sousa Neto[3], Gerbrand Koren[4], Jean Ometto[3], and Mathew Williams[1,2]

[1]School of GeoSciences, University of Edinburgh, Edinburgh UK
[2]National Centre for Earth Observations, University of Edinburgh, UK
[3]INPE, Sao Jose dos Campos, Brazil
[4]Meteorology and Air Quality, Wageningen University, Wageningen, the Netherlands

**Correspondence:** T. L. Smallman (t.l.smallman@ed.ac.uk)

**Abstract.** Identification of terrestrial carbon (C) sources and sinks is critical for understanding the earth system and to mitigate and adapt to climate change results from greenhouse gas emissions. Predicting whether a given location will act as a C source or sink using terrestrial ecosystem models (TEMs) is challenging due to net flux being the difference between far larger, spatially and temporally variable fluxes with large uncertainties. Uncertainty in projections of future dynamics, critical for
policy evaluation, has been determined using multi-TEM intercomparisons, for various emissions scenarios. This approach quantifies structural and forcing errors. However, the role of parameter error within models has not been determined. TEMs typically have defined parameters for specific plant functional types generated from the literature. To ascertain the importance of parameter error in forecasts we present a Bayesian analysis that uses data on historical and current C cycling for Brazil to parameterise five TEMs of varied complexity with a retrieval of model error covariance at 1 degree spatial resolution. After
evaluation against data from 2001-2017, the parameterised models are simulated to 2100 under four climate change scenarios spanning the likely range of climate projections. Using multiple models, each with per pixel parameter ensembles, we partition forecast uncertainties. Parameter uncertainty dominates across most of Brazil when simulating future stock changes in biomass C and dead organic matter (DOM). Uncertainty of simulated biomass change is most strongly correlated with net primary productivity allocation to wood ($NPP_{wood}$) and wood mean residence times ($MRT_{wood}$). Uncertainty of simulated DOM change
is most strongly correlated with $MRT_{soil}$ and $NPP_{wood}$. Due to the coupling between these variables and C stock dynamics being bi-directional we argue that using repeat estimates of woody biomass will provide a valuable constraint needed to refine predictions of the future carbon cycle. Finally, evaluation of our multi-model analysis shows that wood litter contributes substantially to fire emissions necessitating a greater understanding of wood litter C-cycling than is typically considered in large-scale TEMs.

# 1 Introduction

Globally terrestrial ecosystems are estimated to be a net carbon sink sequestering $3.2\pm0.6\,\mathrm{PgC\,yr^{-1}}$ or ~30% of anthropogenic $CO_2$ emissions (Friedlingstein et al., 2019). The net carbon balance of a given ecosystem is dependent on the balance between





larger gross fluxes of uptake by photosynthesis, or gross primary productivity (GPP; 80-170 PgC yr$^{-1}$; Shao et al., 2013; Joiner et al., 2018; Jung et al., 2020), and losses from plant respiration (R$_a$; 40-80 PgC yr$^{-1}$; assuming fixed R$_a$:GPP ratio 0.46; Collalti and Prentice , 2019), heterotrophic decomposition (R$_{het}$; 57.5±9.8 PgC yr$^{-1}$; Sitch et al., 2015) and disturbance such as fire ($\sim$2.2 PgC yr$^{-1}$ (1997-2016); van der Werf et al., 2017). However, the response of terrestrial ecosystems to elevated atmospheric CO$_2$ concentrations and associated climate change are key unknowns in the earth system (Jones et al., 2016). Uncertainties are greatest across the tropics, where data are scarce and process models both diverge in their analysis of current C cycling and exhibit discordant C cycle responses to projected changes in climate (Exbrayat et al., 2019; Shao et al., 2013).

Brazil's ecosystems are among the most biodiverse in the world spanning a range of biomes and climate space (Myers et al., 2000; Lapola et al., 2014): moist-tropical forest of Amazonia and the Atlantic Forests, seasonally dry-tropical grassland forest mosaics of the Cerrado and Caatinga, wetlands in Pantanal and temperate grasslands in Pampa (Figure 1). Brazil's biomes store large quantities of carbon in their biomass and soils, but Brazil is also among the largest emitters of CO$_2$ from land-use change and deforestation (Baccini et al., 2012; Matthews et al., 2014). Between 1990 and 2015 Brazil's forests lost 5.3 PgC (Sanquetta et al., 2018), equating to $\sim$39% of global forest carbon loss for the same period (Köhl et al., 2015). Moreover, the Amazon has been subject to increasingly frequent drought (Lewis et al., 2019) which significantly impacts net carbon exchange due to increased mortality and decomposition (Yang et al., 2018).

Existing process-models of the terrestrial ecosystem simulate carbon stocks that differ significantly from current satellite-based Earth Observation (EO) based estimates and disagree over future trends (Sitch et al., 2008; Huntingford et al., 2013; Shao et al., 2013; Exbrayat et al., 2018a, 2019). Process-orientated terrestrial ecosystem models (TEMs) predict the response of ecosystems to changes in their environment and to disturbance (whether natural or of human origin). Analyses of ensembles of TEM simulations, which are assumed to represent the combined model structural and parameter uncertainty (Todd-Brown et al., 2013; Friend et al., 2014; Jones et al., 2016), have provided valuable information on the likely future dynamics of terrestrial ecosystems (e.g., Friend et al., 2014; Koven et al., 2015; Eyring et al., 2016; Jones et al., 2016; Zhou et al., 2018). However, responses to environmental change are sometimes contradictory indicating model ensemble (i.e. model) specific conclusions (Zhou et al., 2018). For example, using the ISI-MIP ensemble, Friend et al. (2014) showed that on global scales inter-model differences in the mean residence time (MRT) of biomass dominated uncertainty in future carbon stocks, rather than differences in carbon inputs from photosynthesis, while the analysis by Koven et al. (2015), using the CMIP5 ensemble, indicated the reverse. Also using the CMIP5 ensemble, Todd-Brown et al. (2013) showed that while on average simulated soil carbon stocks could be explained by carbon inputs and residence time there was substantial between-model variation as a result of model structural and parameter differences. Lacking a common basis for calibration and evaluation, model-intercomparisons have struggled to identify and reduce uncertainties surrounding model structure and parameterisation.

Ecosystem parameters that drive C exchanges (e.g. plant traits) are known to be highly variable both in space and often in time, even within a given biome (e.g. tropical moist forest) (Butler et al., 2017; Exbrayat et al., 2018b; Kattge et al., 2020). Moreover, field based studies such as those across the Amazon basin have identified substantial spatial variation and trade-off among ecosystem variables including allocation of net primary productivity to plant tissues, MRTs of C pools and carbon



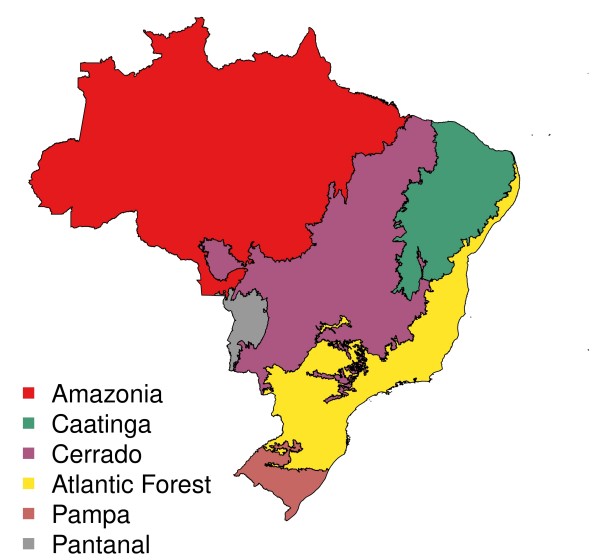

- Amazonia
- Caatinga
- Cerrado
- Atlantic Forest
- Pampa
- Pantanal

**Figure 1.** Map of the major biomes of Brazil. Amazonia and Atlantic forests contain largely moist tropical forest. In contrast the Cerrado and Caatinga are hot and seasonally droughted savannah ecosystems with substantial conversion to agriculture. The Pantanal is covered largely by wetlands. The Pantanal region has a temperate moist climate covered largely by grasslands and agriculture. Map source: Brazilian Institute of Geography and Statistics (IBGE), Biomes and Coastal-Marine System of Brazil map, https://www.ibge.gov.br/, accessed 17/11/2020.

use efficiency (CUE = NPP / GPP) (Doughty et al., 2015; Malhi et al., 2015). In contrast, the majority of TEMs represent ecosystems processes using a limited number of plant functional types (PFT), which assume a single parameterisation for each

biome (i.e. all tropical moist forests are assumed to have the same traits). These TEMs therefore lack spatial variation in the traits that govern the response of ecosystems to changes in their environment, such as disturbance through fire (e.g., Exbrayat et al., 2018b). Furthermore, PFTs are typically calibrated and evaluated at a single site, which may not be representative of a given biomes' mean dynamics (Kuppel et al., 2012).

Model-data fusion (MDF) approaches offer an opportunity to use a diverse array of observations to calibrate and evaluate

TEMs by updating their current state and / or refining their parameters, weighted by observation uncertainty. For example, Exbrayat et al. (2018b) calibrated an intermediate complexity C-cycle model independently in each 1 × 1 degree pixel across the tropics using (among other data) location and time specific information on leaf area index (LAI), biomass and burned area from EO. Their analysis showed substantial within-biome variation in ecosystem variables (and model parameters) in response to varied intensity and frequency of fire. Such variations are neglected in a classical PFT-based TEM framework, introducing

errors into their representation of ecosystem C dynamics. A critical output of a location-based approach is the retrieval of both parameter magnitude and parameter uncertainty information at site level (i.e. pixel or grid cell). Such information can highlight





the greatest unknowns, underpin explicit uncertainty propagation into future environments (e.g. climate change), and directly investigate the parameter-process uncertainty interactions that drive changes in ecosystem C stocks.

Uncertainty in future carbon cycle simulations (e.g., Arora et al., 2020) is dominated by combined model structure and parameter uncertainty, however there remains a substantial contribution due to variations between climate change scenarios themselves (Lovenduski and Bonan , 2017; Bonan and Doney, 2018). The potential mean global warming is estimated to be 1.7 - 5$^o$C by 2100 (IPCC 2014). This uncertainty is driven largely by broad ranges in anthropogenic emissions and land-use and land-cover change for which there are many plausible pathways leading to different atmospheric $CO_2$ concentrations (O'Neill et al., 2016) for use in model intercomparisons (e.g., Eyring et al., 2016). For example, Bonan et al. (2019) showed that both land use and climate change scenarios had a significant impact on TEM-simulated terrestrial carbon stocks, indicating the need to include future scenarios in any uncertainty partitioning experiment.

Here we use the CARbon DAta MOdel fraMework (CARDAMOM; Bloom et al., 2016) to calibrate a suite of five intermediate complexity TEMs across Brazil (1 × 1 degree pixel; 2001-2017; monthly time step) to retrieve ensembles of pixel-specific parameters. These localised parameter ensembles provide explicit estimates of parameter uncertainty and its spatial variability. The five models have a common basic structure, but progressively more complex process representation, allowing quantification of the model ensembles' structural uncertainty. Moreover, our approach mimics the typical TEM model development process where incremental changes in process representation are evaluated for their impact on simulated outcomes (e.g. Mercado et al., 2009; Verheijen et al., 2015; Jones et al., 2020) while we go further by explicitly quantifying the associated parameteric uncertainty which is usually unavailable. Comparison of the dynamics of this model ensemble against independent estimates of the Brazilian C cycle quantifies whether a given change in model structure (i.e. added complexity) leads to an improvement, degradation or equally valid C cycle analysis. Once calibrated we simulate each model for each pixel over a parameter ensemble and under multiple climate change scenarios providing quantification of climate scenario uncertainty to 2100.

Using this approach we address the following research questions:

1) Does increasing model complexity improve agreement with independent evaluation information? Firstly, we hypothesise that including a water-cycling sub-model will reduce photosynthesis due to soil moisture limitations and so improve model outputs for the drier Cerrado and Caatinga regions. Secondly, we hypothesise that inclusion of a wood litter pool will increase fire emissions by adding another combustible dead organic matter pool and improve estimated emissions particularly within areas of forest cover loss such as the south eastern edge of the Amazon (the arc of deforestation).

2) How is uncertainty associated with predicted carbon stocks partitioned between (i) parameter estimates, (ii) model structure and (iii) projected climate change scenario? We hypothesise that the climate change scenario will contribute a minor component to the overall uncertainty, consistent with the results found by Bonan et al. (2019). Additionally, we hypothesise that parameter uncertainty will be largest in areas of large biomass and soil carbon stocks (i.e. Amazon and Atlantic Forests) due to larger uncertainties found in observational constraints at larger values whereas model structure uncertainty will be more important in regions with lower stocks and more seasonality in fluxes, such as Cerrado.





3) Is forecast uncertainty more strongly linked to the mean residence time (MRT) of biomass or differences in carbon inputs from photosynthesis? How does the relative importance of these factors vary spatially across biomes and among models with different process representation? We hypothesise that biomes with stronger environmental constraints on production, for instance dry tropics compared to moist tropics, will have errors dominated more by this process than MRT.

We investigate these questions at the scale of Brazil and also for its key biomes, to determine if there are regional differences. We conclude with an assessment of key steps required to produce more robust projections of how Brazilian C stocks will respond to future forcing. The novelty of this study is to compare C cycle projections that include propagated error from model calibration at pixel scale, for a range of models with difference process controls on C cycling, allowing a robust data-constrained analysis.

## 2 Methods

We use the CARbon DAta MOdel fraMework (CARDAMOM; Bloom et al., 2016) to perform a model-data fusion (MDF) analysis of Brazil at $1 \times 1$ degree spatial and monthly temporal resolutions between 2001 and 2017. CARDAMOM retrieves ensembles of model parameters independently for each location (see Sec. 2.1) as a function of location specific observational constraints (see Sec. 2.3). To quantify model structural uncertainty, parameters are retrieved for five versions of the DALEC
terrestrial ecosystem model, of differing complexity (see Sec. 2.2; Table 1). The DALEC calibrations are evaluated using independent estimates of net ecosystem exchange of $CO_2$ (NEE), GPP and Fire (see Sec. 2.4). The calibrated DALEC models are then simulated into the future under four climate change scenarios (see Sec. 2.5) used in Phase 6 of the Coupled Model Intercomparison Project (CMIP6; O'Neill et al., 2016).

**Table 1.** Summary information of process representations and total number of calibrated parameters of the different DALEC models. Model complexity increases from model M1-5. ACM is the Aggregated Canopy Model used to predict photosynthesis (GPP), with ACM1 the simplest version and ACM2 more complex with links to water balance. Plant respiration ($R_a$) can be determined as a simple ratio of GPP or by separate maintenance ($R_m$) and growth ($R_g$) components.

| Model | Photosynthesis | Water cycle | Plant respiration | Wood litter | No. parameters |
|-------|----------------|-------------|-------------------|-------------|----------------|
| M1 | ACM1 | No | $R_a$:GPP | No | 23 |
| M2 | ACM2 | No | $R_a$:GPP | No | 23 |
| M3 | ACM2 | Yes | $R_a$:GPP | No | 23 |
| M4 | ACM2 | Yes | $R_m$:GPP + $R_g$:NPP | No | 27 |
| M5 | ACM2 | Yes | $R_m$:GPP + $R_g$:NPP | Yes | 29 |

### 2.1 CARDAMOM

CARDAMOM is a MDF framework which uses a Bayesian approach within an Adaptive Proposal - Markov Chain Monte Carlo (AP-MCMC) to estimate ensembles of DALEC model parameters (x; Table A1) consistent with observational constraints and





their uncertainties (Haario et al., 2001; Rodríguez-Veiga et al., 2020). CARDAMOM analyses are conducted independently for each pixel location and repeated three times (each repeat is known as a chain). Each chain assesses 100 million parameter proposals from which a sub-sample of 1000 accepted parameter vectors are stored for post-processing. The chains are used to assess AP-MCMC quality; in each location the chains are expected to statistically converge based on the Gelman-Rubin convergence criterion (Gelman and Rubin , 1992). Any location which did not achieve convergence is re-run. For further details see SI text S1.1.

## 2.2 DALEC models

The DALEC model suite used here comprises five related intermediate complexity models of the terrestrial carbon cycle (M1-5). Each model version tracks the state and dynamics of live and dead carbon pools, their interactions, and their response to climate and exogenous factors such as fire or disturbance. The complexity (numbers of carbon pools, their connectivity) and process representation (component sub-models of varying complexity) varies between DALEC models. There are three alternate carbon cycle structures (Figure 2), plus a range of different sub-models (Table 1). The sub-models are related to different simulations of GPP, $R_a$, and carbon-water interactions. These sub-models build on a common baseline structure, facilitating efforts to disentangle the impact of each specific process representation. Due to their varied complexity the DALEC models have different numbers of parameters which are calibrated for each location. DALEC parameters for each model can be found in Table A1 including a summary of the key features of each model in SI text S1.2.

## 2.3 Observational constraints and driving information

CARDAMOM uses a diverse array of data as both observational constraint and model inputs. Information on LAI (time series), above ground biomass (AGB) and soil carbon are assimilated observations with an associated uncertainty. Meteorology, burned area, forest cover loss and soil texture (sand / clay fractions) are inputs without uncertainty. Summary information on the assimilated observations and their uncertainties shown in Figure A1.

### 2.3.1 Leaf area index

Time series information on LAI magnitude and uncertainty is extracted from the $1 \times 1$ km, 8-day product from the Copernicus Service Information (2020). LAI was aggregated to the analysis resolution. Each LAI estimate has a corresponding uncertainty value, however the robustness of the uncertainty provided with EO LAI products remains unclear (Zhao et al., 2020). To be conservative we assumed the maximum uncertainty value reported from the raw data used in the aggregation of each time step. Each pixel will typically assimilate 204 EO-based LAI estimates, i.e. 12 months $\times$ 17 years.

### 2.3.2 Wood C

A single estimate per-pixel of AGB and uncertainty is extracted from a combination of the Avitabile et al. (2016) and Longo et al. (2016) maps. Avitabile et al. (2016) combines multiple years of EO and field data to create a pan-tropical map nominally



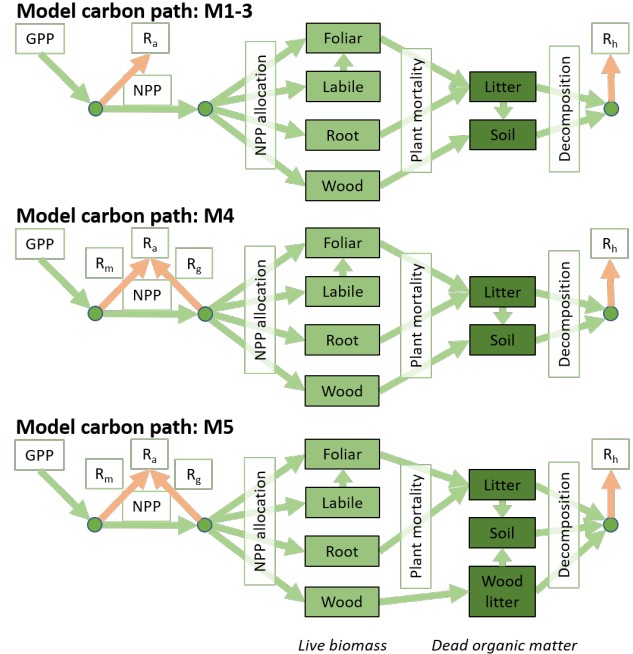

**Figure 2.** The five DALEC versions include three carbon cycle structures. The top schematic shows the DALEC model carbon structure as previously described in Bloom & Williams (2015) as is used for M1-3. The middle row shows M4 where $R_a$ is partitioned between $R_m$ and $R_g$. The bottom row shows the inclusions of a wood litter pool used in M5.

representative of 2007. Longo et al. (2016) covers the Brazilian Amazonia only and uses field inventory and airborne lidar explicitly representing 2014. These maps are created using different source data and algorithms; they will contain unique errors and bias. Therefore, we use the Longo et al. (2016) map to provide constraint on Amazonian AGB and the Avitabile
et al. (2016) map elsewhere (Figure A1).

The DALEC models simulate a combined above and below ground woody pool. To link the AGB maps to the simulated wood pool we use an allometric relationship to estimate the below ground biomass (BGB) following Saatchi et al. (2011) (units = Mg/ha)

$$BGB = 0.489 \cdot AGB^{0.89} \tag{1}$$

AGB uncertainty is similarly converted based on the allometric equation neglecting statistical uncertainty of the allometric equation itself. Wood C and its uncertainty are then spatially aggregated assuming uncorrelated uncertainties as error covariance remains unknown.





### 2.3.3 Soil C and texture

Location specific estimates of soil carbon and sand / clay fraction are extracted from the SoilGrids database (Hengl et al.,
2017). Soil carbon is used as a prior on the initial soil carbon stock while soil texture information is used as an input to the soil
hydrology sub-model. SoilGrids uses inventory data of soil properties and interpolates these across a $250 \times 250$ m grid using a
machine learning (ML) approach (Hengl et al., 2017). However, SoilGrids lacks an estimate of uncertainty. For simplicity, we
assumed an uncertainty was the standard deviation of the spatially aggregated dataset.

### 2.3.4 Disturbance

Fire and forest biomass removal was imposed using EO information. The MODIS burned fraction product (Giglio et al., 2018)
determines the areas where fire is imposed. Emissions are determined assuming a fraction of simulated biomass is combusted or
converted to litter based on tissue specific combustion-completeness factors, following Exbrayat et al. (2018b). Forest biomass
removal is imposed using the global forest watch (GFW) forest cover loss product (Hansen et al., 2013). GFW provides the
year in which a forest area is removed, with biomass losses assumed to occur evenly across the year. All biomass is assumed
to be subject to removal except fine roots which remain in the ecosystem

### 2.3.5 Meteorological drivers

Meteorological drivers are drawn from the CRU-JRAv1.1 dataset, a 6-hourly $0.5 \times 0.5$ degree reanalysis (CRU , 2019). At-
mospheric $CO_2$ concentration is taken from the Mauna Loa global $CO_2$ concentration (www.esrl.noaa.gov/gmd/ccgg/trends/,
accessed: 22/08/2020). Owing to their differing complexities the DALEC models use different drivers. All models use temper-
ature, shortwave radiation and atmospheric $CO_2$ concentrations. M2-5 additionally use vapour pressure deficit and wind speed.
M3-5 use precipitation. Summary information for mean climate is shown in Figure A2.

## 2.4 Evaluation of models against independent data

To address research question 1, the five models were evaluated against a series of independent data on Net Ecosystem Exchange
of $CO_2$ (NEE), gross primary productivity (GPP) and fire emissions. These datasets were derived from atmospheric inversions,
upscaling from flux measurements, and from remote sensing of burned area.

   CarbonTracker Europe (CTE) is a widely used atmospheric inversion system which estimates NEE by combining time
varying prior information on NEE along with imposed $CO_2$ fluxes from fire, fossil fuels and ocean exchange with observations
of atmospheric $CO_2$ concentrations (van der Laan-Luijkx et al., 2017). A single CTE analysis which spans the whole analysis
period (2001-2017) is used to provide a long-term comparison of the trend in NEE (van der Laan-Luijkx et al., 2017). Spatial
comparisons are restricted to the 2009-2017 period using the dataset described below.

   An ensemble of 15 analyses ($1 \times 1$ degree; 2009-2017) which builds on the CarbonTracker South America (CT-SAM)
framework provided robust uncertainty estimates of NEE. In addition to CTE's standard atmospheric measurements CT-SAM
includes airborne estimates focused over the Amazon forest (Gatti et al., 2014), and uses zoom regions over South America for





improved atmospheric transport (van der Laan-Luijkx et al., 2015). The ensemble uses five NEE priors and three fire emission

drivers but with a common set of atmospheric constraints and transport model (Schaefer et al., 2008; Bodesheim et al., 2018; van Schaik et al., 2018; Haynes et al., 2019; Koren , 2020). By using a range of priors it covers the uncertainty in the seasonal variation of C fluxes in tropical regions (Saleska et al., 2003; Restrepo-Coupe et al., 2013; Koren et al., 2018; Mengistu et al., 2020). In the remainder of this text the CTE and CT-SAM datasets are collectively referred to as CTE.

FLUXCOM GPP is estimated by an ensemble of ML approaches driven with meteorological reanalysis and EO derived

vegetation indices and calibrated using eddy covariance information drawn from the FLUXNET network (Jung et al., 2020). FLUXCOM has been widely evaluated using eddy covariance information and has been used to evaluate TEMs (Jung et al., 2020). Each GPP estimate has a corresponding median absolute deviation estimate drawn from the ML ensemble which is assumed to represent FLUXCOM's uncertainty for comparisons to CARDAMOM-DALEC.

Independent estimates of fire emissions are drawn from the Global Fire Emissions Database version 4.1s (GFEDv4.1s

(2001-2017); van der Werf et al., 2017) and the Global Fire Assimilation System (GFAS (2003-2017); Kaiser et al., 2012). Neither product comes with uncertainty information. GFEDv4.1s uses MODIS burned area to impose fire on a TEM with actual emissions determined based on the simulated magnitude of carbon pools at steady state in conjunction with pool specific combustion completeness parameters (van der Werf et al., 2017). The GFAS product uses MODIS radiative energy and active fire products combined with ecosystem specific parameters to estimate carbon emissions from fire (Kaiser et al., 2012). As

these products are based on fundamentally different approaches we assume that the range between them approximates the fire emissions uncertainty. We evaluate DALEC for their overlapping period (2003-2017).

### 2.5 Analysing the drivers of forecast uncertainty

To project DALEC to 2100, future climate drivers were extracted from the UK Earth System Model (UKESM; Sellar et al., 2019) contribution to CMIP6 (Eyring et al., 2016). This study uses the core scenarios SSP1-2.6W m$^{-2}$, SSP2-4.5W m$^{-2}$,

SSP3-7.0W m$^{-2}$, SSP5-8.5W m$^{-2}$ spanning a mean global warming of 1.7-5$^{o}$C (O'Neill et al., 2016). The scenarios are also used to impose future forest biomass extraction. The future climate is imposed by determining the anomaly from the end of our analysis until 2100. The mean temperature (M1-5), incoming shortwave radiation (M1-5), vapour pressure deficit (M2-5), wind speed (M2-5) and precipitation (M2-5) anomalies for each scenario as shown in Figure A3. The time series of future atmospheric CO$_2$ concentration is prescribed for each scenario.

Model analyses quantified the relative contribution of variation in model parameters, model structure and climate change scenario to overall uncertainty in the simulation of biomass and DOM to 2100. Parameter uncertainty was estimated to be the 90% CI resulting from the simulation of the retrieved parameter ensembles. Model structural uncertainty was estimated as the between model range of the pixel-level median estimates. Climate change scenario uncertainty was estimated as the pixel-level range of median estimates across scenarios for each model. This analysis address research question 2.

To address question 3, quantifying the role of key ecosystem traits (NPP partitioning and MRTs) on C stock trajectories, the ensemble of pixel level estimates of GPP, CUE, NPP allocation and MRTs are correlated with the ensemble of biomass and DOM stock change estimated between 2001-2100. It is the ensembles of per-pixel parameters and by extension ensembles





of C stock and flux estimates, that uniquely allow CARDAMOM to explicitly quantify the uncertainty in critical ecosystem properties with C stock dynamics.

**Table 2.** Summary of Brazil wide carbon budgets for each DALEC model and independent estimates. Fluxes are gross primary productivity (GPP), autotrophic respiration ($R_a$), heterotrophic respiration ($R_h$), forest biomass loss, carbon emissions due to fire, net ecosystem exchange of $CO_2$ (NEE = $R_a$ + $R_h$ - GPP) and net biome exchange (NBE = NEE + Fire) . All units are in PgC yr$^{-1}$. Values given as the median pixel level estimates averaged across Brazil while values in parenthesis are Brazil wide averaged for the 5% and 95% quantiles. Independent estimates are derived from FLUXCOM (FC, for GPP), Global Forest Watch (GFW, for forest loss), Global Fire Emissions database v4.1s (GFEDv4.1s, for fire) and CarbonTracker Europe (CTE, for NEE). Time period of all data is 2001-2017.

| Flux | M1 | M2 | M3 | M4 | M5 | Independent estimate |
|---|---|---|---|---|---|---|
| GPP | 17.7 (9.8 / 23.4) | 19.0 (14.6 / 22.3) | 17.8 (12.7 / 21.8) | 17.8 (12.7 / 21.8) | 17.8 (12.7 / 21.8) | 18.4 |
| $R_a$ | 7.7 (3.5 / 12.9) | 8.8 (5.1 / 13.0) | 7.8 (4.2 / 12) | 7.7 (3.8 / 11.9) | 7.6 (3.8 / 11.9) | - |
| $R_h$ | 7.3 (3.2 / 14.3) | 7.4 (3.6 / 14.1) | 7.6 (3.7 / 14.2) | 7.6 (3.8 / 14.4) | 7.3 (3.5 / 14.0) | - |
| Fire | 0.11 (0.07 / 0.19) | 0.11 (0.08 / 0.18) | 0.11 (0.07 / 0.18) | 0.12 (0.08 / 0.23) | 0.17 (0.1 / 0.38) | 0.20 |
| Forest loss | 0.18 (0.12 / 0.28) | 0.18 (0.12 / 0.28) | 0.18 (0.12 / 0.29) | 0.18 (0.12 / 0.28) | 0.18 (0.13 / 0.28) | 0.24 |
| NEE | -1.9 (-6.1 / 3.9) | -2.3 (-6.2 / 3.5) | -2.0 (-6.3 / 4.4) | -2.1 (-6.4 / 4.5) | -2.5 (-6.7 / 3.9) | -0.26 |
| NBE | -1.8 (-5.9 / 4.0) | -2.2 (-6.1 / 3.7) | -1.9 (-6.1 / 4.6) | -2.0 (-6.3 / 4.7) | -2.3 (-6.5 / 4.1) | - |

## 3 Results

We conducted MDF analyses to retrieve ensembles of location specific parameters for five DALEC models of varied complexity across Brazil. Each model simulated the calibration data with a good degree of skill, returning similar likelihood scores (R2 >0.98; Figures 3-5, S4-5). Simulated NEE, GPP and fire emissions have been evaluated at 1 degree (Figure 4) and national scale (Figure 5; Table 2) against independent estimates. The 1 degree spatial parameter ensembles show that there is a strong dependency between net carbon exchange and wood stocks (R2 >0.8; Figure A6). The DALEC parameter ensembles have been projected to 2100 (Figure 6) and show that parameter, not structure or climate change scenario, dominates overall uncertainty in most areas (Figure 7). Finally, using the parameter ensembles to quantify the correlation between ecosystem variables and future carbon stock dynamics we identify allocation of NPP to wood (NPP$_{wood}$) and MRT$_{wood}$ as targets for further constraint on model forecasts (Figures 8-9).

## 3.1 Calibration constraints

All DALEC models match their calibration information with a high degree of skill (Figure 3). The root mean square error (RMSE) is small for LAI and the initial soil carbon stock (<5 %). RMSE between simulated wood stocks and calibration observations is larger in wood stocks (<16 %) and is dominated by model-observation mismatch at smaller wood stocks (<50 MgC/ha; 20-28 %) with smaller errors (<1 %) otherwise. The calibrated Ra:GPP (CUE = 1-Ra:GPP) fraction across Brazil is consistent with the assimilated prior (M1,4,5 = 0.43; M2 = 0.46; M3 = 0.44) for each model (mean deviation <0.06; prior =





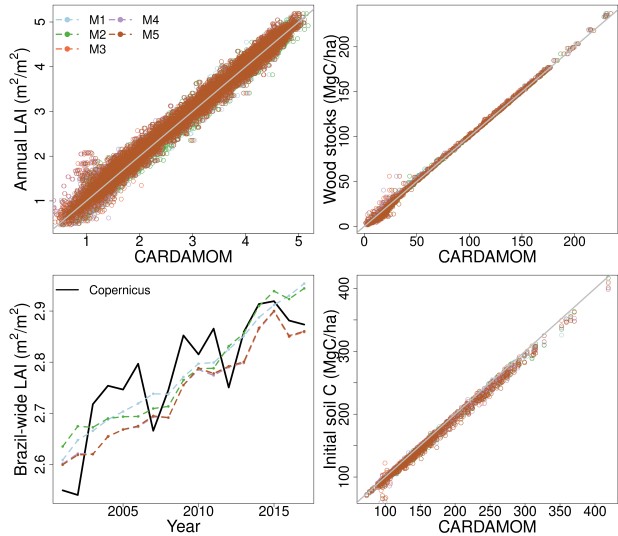

**Figure 3.** Comparison between observational constraints used to calibrate the DALEC models and the corresponding model outputs. The grey line shows the 1:1 line. LAI is presented as a pixel wise comparison of the mean annual LAI and as a Brazil-wide average time series highlighting the long term trend of increasing LAI.

0.46+/-0.12; Figure A5). The largest deviations from the prior are found in Caatinga (Figure A5), a hot, dry biome with mean air temperature >25 $^o$C and rainfall <800 mm yr$^{-1}$ (Figure A2). In M2 (ACM2, no water cycle) Caatinga has a higher Ra:GPP (deviations up to 0.33) while models M3-5 (ACM2, with water cycle) estimate a lower Ra:GPP in Caatinga (deviations up to -0.25). M1 shows no substantial spatial patterning (ACM1, no water cycle). The inter-model differences follow the switching between photosynthesis models and the inclusion of the water cycle, indicating that drought stress has a significant impact on Ra:GPP (Table 1).

### 3.2 Independent evaluation of Brazilian C-cycling

DALEC-simulated NEE was statistically consistent with the CTE ensemble at the 90% confidence interval (CI) across >99% of Brazil (2009-2017; Figure 4), i.e. there is overlap between CARDAMOM's 90% CI and the spread of estimates from the CTE ensemble. Moreover, both CTE and DALEC models indicate a long-term decreasing trend in NEE (i.e. increasing net carbon uptake) (Figure 5). While statistically consistent there is persistent negative bias between DALEC and CTE (i.e. DALEC models estimate a greater C sink than CTE) over the Amazon (M1-5; Figure A7). This Amazon bias leads to the DALEC models consistently estimating Brazil's NEE to range between -2.5 and -1.9 PgC yr$^{-1}$, a larger sink than the -0.26 PgC yr$^{-1}$ estimated by CTE (2001-2017; Table 2). However, both the CTE ensemble and DALEC uncertainties cross the source / sink boundary, indicating that neither analysis can confidently quantify Brazil as a net source or sink overall for 2009-2017 (Figure 5; Table 2).



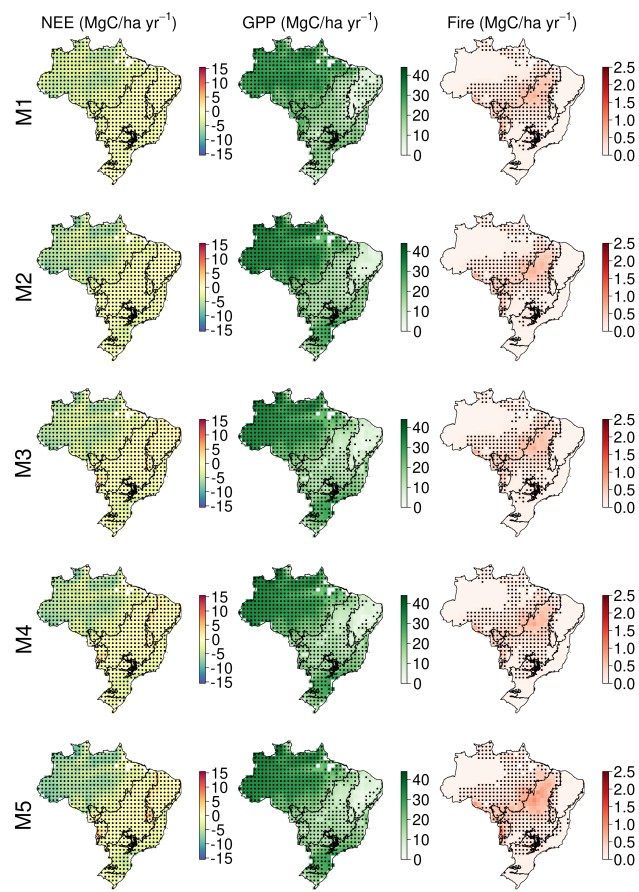

**Figure 4.** Comparison of DALEC model M1-5 estimated NEE and GPP with independent estimates of NEE from CarbonTracker Europe (2009-2017), GPP (2001-2017) from FLUXCOM and fire (2003-2016) GFEDv4.1s & GFAS. Stippling shows areas where the 90% confidence interval derived from DALEC analysis overlaps the independent value.

The DALEC models are consistent with FLUXCOM GPP at the 90% CI across 83-94% of Brazil (Figure 4). Inter-model variation follows the implementation of the differing photosynthesis models and inclusion of carbon-water cycle interactions. The simplest model, M1, (ACM1, no water cycle; 94%) was the most consistent with FLUXCOM; followed by M2 (ACM2, no water cycle; 90%). M3-5, which use ACM2 and simulate water cycling, are least consistent with FLUXCOM (83-85%) over Brazil. The non-consistent areas for all models are concentrated in the Caatinga and Cerrado (Figure 4), which have strong seasonality in rainfall and more extreme temperatures (Figure A2). Moreover, the DALEC models all estimate a lower GPP for these regions (by $\sim$5 MgC/ha yr$^{-1}$) than FLUXCOM suggesting different high temperature and drought sensitivities between analyses (Figure A7). The activation of water cycling between M2 and M3-5 reduces Brazil's GPP by an average of 7% but varies substantially in space with declines across the Cerrado and Caatinga of $\sim$30%.

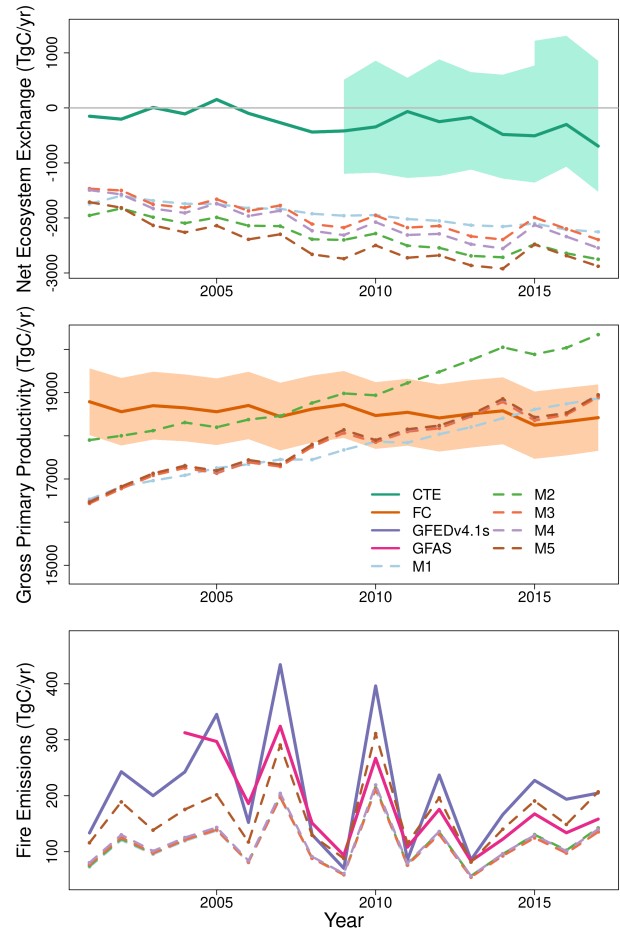

**Figure 5.** Time series comparison showing the carbon budget for Brazil as estimated by CARDAMOM for each version of DALEC using the pixel level median estimates. Independent data, where available, is shown as a single solid line. The shaded areas denote the maximum range the independent estimate for CTE (blue) and the median absolute deviation for FLUXCOM (orange).

Brazil's mean annual GPP estimated by the DALEC models (17.7 - 19.0 PgC yr$^{-1}$) encompasses FLUXCOM's (18.4 PgC yr$^{-1}$) as do the model specific uncertainties (Table 2). All DALEC models estimated Brazil's mean annual GPP to increase between 2001 and 2017 ($\sim$0.15 MgC/ha yr$^{-2}$) while respiration increased by roughly half that of GPP (Figure 5, S8). The net change between GPP and respiration is consistent with the trend of declining NEE estimated by both DALEC (-0.038 to

-0.06 MgC/ha yr$^{-2}$) and CTE (-0.032 MgC/ha yr$^{-2}$; Figure 5). However, the increasing GPP trend is not uniform, with GPP declining in the Cerrado and Caatinga (range $\pm$0.45 MgC/ha yr$^{-2}$; Figure A8). In contrast, FLUXCOM shows a net decreasing trend between 2001 and 2017 (mean = -0.015 MgC/ha yr$^{-2}$; range = -0.092 to 0.032 MgC/ha yr$^{-2}$; Figure A9). But the largest FLUXCOM estimated declines are also across the Cerrado and Caatinga with increases in the south of Brazil and the Atlantic forest.





DALEC-estimated fire emissions show large inter-annual variation between ~100 TgC yr$^{-1}$ and ~300 TgC yr$^{-1}$, reducing the annual net C uptake by 3-30 % over this period (Figure 5). At the 90% CI the DALEC models are consistent with GFEDv4.1s and GFAS fire emissions estimates over 41-53% of Brazil (Figure 4). DALEC has, however, a persistent low bias (~2 MgC/ha yr$^{-1}$) across the boundary between the Amazon and Cerrado (Figure A7). Spatial consistency is greatest in M5 (53%) and M4 (49%). Moreover, a comparison of total fire emissions show that M1-4 estimates are substantially lower than

either GFAS (mean = 65-69 TgC yr$^{-1}$) or GFEDv4.1s (mean = 92-96 TgC yr$^{-1}$) (Figure 5). Whereas M5 estimates, while still lower than GFAS (mean = 13 TgC yr$^{-1}$) or GFEDv4.1s (mean = 41 TgC yr$^{-1}$), fall between the independent estimates in 11 of 15 years in which these data sets overlap (Figure 5). Collectively, these results suggest that increasing model complexity by both partitioning R$_g$ and R$_m$, and the inclusion of a wood litter pool, improved simulation of fire (Table 1). DALEC-estimated C losses due to forest biomass removals from GFW show substantial inter-annual variation (120-400 TgC yr$^{-1}$) reducing net

uptake by 5-32 % (Figure A10). For further details see SI text S2.

**Table 3.** Mean Brazilian C cycle NPP allocation fractions and mean residence times (MRT; years) calculated with five different model structures (M1-5). The Brazil-wide mean is calculated from the median pixel level estimates, with the equivalent estimates for the 5 % and 95 % quantiles in parenthesis denoting the 90 % confidence interval. Note that litter in M5 includes the additional wood litter pool.

|  |  | M1 | M2 | M3 | M4 | M5 |
|---|---|---|---|---|---|---|
| NPP | Foliage | 0.22 (0.09 / 0.41) | 0.21 (0.1 / 0.38) | 0.21 (0.1 / 0.38) | 0.18 (0.09 / 0.32) | 0.19 (0.10 / 0.33) |
|  | Fine root | 0.47 (0.20 / 0.73) | 0.52 (0.26 / 0.74) | 0.48 (0.22 / 0.71) | 0.46 (0.21 / 0.69) | 0.43 (0.19 / 0.67) |
|  | Wood | 0.28 (0.1 / 0.55) | 0.25 (0.1 / 0.49) | 0.29 (0.1 / 0.54) | 0.28 (0.1 / 0.53) | 0.30 (0.1 / 0.54) |
| MRT | Foliage | 1.3 (0.6 / 2.3) | 1.5 (0.8 / 2.4) | 1.5 (0.8 / 2.4) | 1.6 (0.9 / 2.4) | 1.5 (0.8 / 2.4) |
|  | Fine root | 0.48 (0.19 / 1.7) | 1.0 (0.39 / 1.85) | 0.86 (0.3 / 1.8) | 0.82 (0.29 / 1.8) | 0.80 (0.29 / 1.8) |
|  | Wood | 16.3 (5.6 / 58.7) | 15.2 (5.5 / 51.8) | 15.5 (5.8 / 50.5) | 14.9 (5.6 / 49.3) | 21 (7.0 / 58.6) |
|  | Litter | 0.18 (0.08 / 1.5) | 0.18 (0.08 / 1.4) | 0.18 (0.07 / 1.3) | 0.24 (0.1 / 2.8) | 0.96 (0.29 / 3.4) |
|  | Soil | 36.0 (10.3 / 146) | 36.2 (11.0 / 131.3) | 33.2 (10.3 / 120) | 27.8 (9.5 / 73.6) | 38.4 (12.2 / 110.3) |

### 3.3   Constraints on Brazilian C-cycling

Simulated net biome exchange (NBE = NEE + fire) is dominated by wood stock dynamics. Variation in wood stock dynamics explains 81-92 % of variation of simulated NBE while variation in soil stock dynamics explains 2-16 % (Figure A6). Carbon dynamics of wood, not soil, is the primary driver of net exchange. Using the per-model, 1 degree resolution parameter ensem-

bles provides quantification across Brazil of whether a given 1 degree pixel is a net source or sink of carbon, i.e. the sign of NBE (Figure A6). At the 90 % confidence interval our analyses indicate there is currently insufficient observational constraint to confidently determine the sign of NBE or soil C dynamics. The same was largely true for wood stock dynamics, except that the sign of wood stock trajectories could be confidently determined for ~5 % of Brazil's land area after the inclusion of the water cycle increasing to 11% on the inclusion of a wood litter pool. The areas of statistical confidence are concentrated in the

Cerrado (Figure A6).





The CARDAMOM analyses provide 1 degree spatial resolution estimates of critical ecosystem traits such as the allocation of NPP to live tissues and carbon stock MRT (Table 3). All five models estimate similar mean Brazil-wide partitioning of NPP to plant tissues with ∼ 20 % allocated to foliage, 30 % to wood and the majority ∼50 % to fine roots. The MRT of the foliar pool is best constrained across models, consistently estimated to be ∼1.5 years with an uncertainty range of 116 days in M1 decreasing to 84 days in M5. The MRT of fine roots increases by 67-101% between M1 which uses ACM1 and M2-5 which use ACM2. A key feature of ACM2 is the inclusion of fine roots in determining potential water supply to the canopy underpinning stomatal conductance. Wood MRT while associated with large uncertainty (∼50 years) is consistently estimated to be 15-16 years for M1-4. In M5 with the inclusion of the wood litter pool increased median wood MRT by 25 % to 21 years. No significant impacts on mean estimates or uncertainty of litter or soil MRT are noted.

These analyses show clear spatial patterning both between and within biome in the estimates of NPP (Figure A11) and MRT (Figure A12). Spatial patterns of NPP allocation are similar between models, except M2, which has several notable differences in the Cerrado and Caatinga. All models estimate the fraction of NPP allocated to foliage across the Amazon, Atlantic forest, Pantanal and Pampas to be relatively low at 0.1-0.2 with a larger fractional allocation (0.3-0.5) estimated across the Cerrado and Caatinga. Models M1,3-5 estimate the fractional allocation to fine roots across Brazil to be 0.5-0.6 except in the Cerrado and Caatinga which have a lower fractional allocation at ∼0.4. These models also estimate a broadly similar allocation fraction of NPP to wood across Brazil (0.25-0.30) with marginally higher allocation (0.3-0.35) across the Cerrado and arc of deforestation (Amazon - Cerrado boundary). M2 estimated a similar spatial pattern and magnitude of fine root allocation except over the Cerrado and especially the Caatinga which is estimated to have a larger allocation fraction of up to 0.8. The larger fractional allocation to fine roots comes as a trade-off with allocation to wood leading to very low wood allocation fractions across the Cerrado and Caatinga (< 0.15).

There are substantial variations between models in estimated MRTs, in addition to biome level differences (Figure A12). $MRT_{root}$ shows the greatest between-model variation with short (<1 year) MRTs estimates across the majority of Brazil (i.e. little biome level variation) in M1 but longer MRT in all other models (1-2 years). Models M2-5 have larger biome-level variability in MRT, with longer (> 1 year) $MRT_{root}$ in the drier Cerrado and Caatinga regions compared to other biomes. In models M1-4 $MRT_{wood}$ is ∼10 years across much of Brazil except the Amazon which has MRTs of up to 50 years but notable short MRT along the boundary of with the Cerrado (i.e. the arc of deforestation). Longer residence times were estimated across parts of the Caatinga in M3-5, likely linked to the inclusion of the water cycle in these models. In M5, where an explicit wood litter pool is included, $MRT_{wood}$ increased in the southern Cerrado and Atlantic forest from < 10 years to > 10 years. Finally, the estimate of the combined litter (i.e. foliar, fine root and wood) MRT in model M5 was greater across Brazil than other models due to the explicit inclusion of slowly decomposing wood litter.





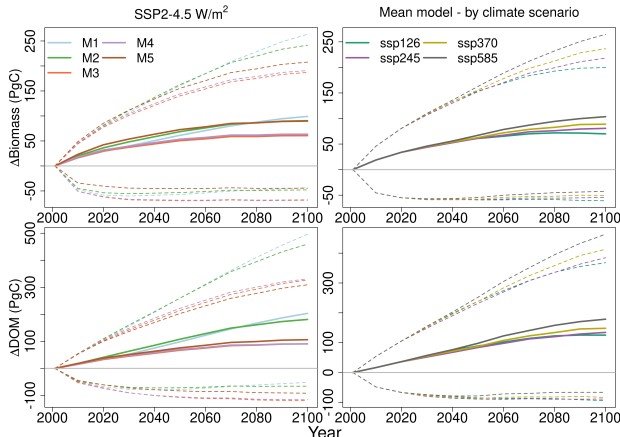

**Figure 6.** Simulated Brazil-wide total stock change between 2001 and 2100 for Biomass (labile + foliar + fine root + wood) and dead organic matter (DOM; Litter + Wood Litter (M5 only) + Soil). The left column shows each model for the SSP2-4.5 W m$^{-2}$ climate scenario while the right column shows the DALEC suite mean for each climate scenario. Median estimates are shown using a solid line while the dashed lines indicate the 90% confidence interval.

## 3.4 Quantifying uncertainty in the future Brazilian C-cycle

### 3.4.1 Future Brazilian carbon stocks

The calibrated DALEC models were simulated under four climate change scenarios to estimate changes in biomass and DOM between 2001 and 2100 (Figure 6). We assess the model specific behaviour under scenario SSP2-4.5 W/m2, which is considered to be the central pathway, and the model average response for each of the four scenarios (O'Neill et al., 2016).

The median forecasts of all DALEC models simulated a net increase of biomass and DOM by 2100 under the SSP2-4.5 W m$^{-2}$ climate change scenario (Figure 6, S3). M1-2 and M5 simulated a larger C accumulation ($\sim$75 PgC) while M3-4 simulate a smaller increase ($\sim$40 PgC). The 90% CI (i.e. the 5% - 95% quantiles) is greater than the median predicted accumulation for each model, therefore crossing the source / sink boundary for the next century in all cases (Figure 6). C accumulation is simulated to be concentrated in the Amazon and to a lesser degree the Atlantic forests (e.g. M5). All other areas (i.e. the drier regions) remain near neutral or decline in carbon storage (Figure A13, A14).

The analyses indicate that live biomass and DOM stocks will most likely increase under each climate change scenario (Figure 6). Median C accumulation under SSP1-2.6 W m$^{-2}$ plateaus by 2080 before turning into a carbon source by 2100 (i.e. begins losing its accumulated carbon), while all other scenarios continue to accumulate carbon to 2100. As expected, accumulation of DOM lags that of biomass, as turnover of biomass provides inputs to DOM in the models. Analysis uncertainty is larger than mean predicted accumulation for all scenarios, with the lower bound of the 90% CI indicating a possible net loss of carbon for the next century. The spatial variation in carbon source / sink distribution indicated for SSP2-4.5 W m$^{-2}$ is consistent for each scenario (Figure A13, A14).








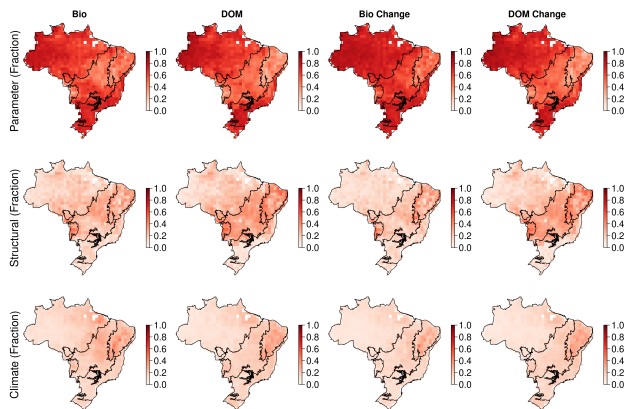

**Figure 7.** Quantification of the relative contribution of model parameter (top row), structural (middle row) and climate change scenario (lower row) uncertainties on the simulated Brazilian carbon cycle in 2100 at 1 degree resolution. Carbon cycle components shown in columns as the live biomass (Bio) and dead organic matter (DOM) stocks in 2100, and their change between 2017 and 2100. The colour bar indicates the fractional contributions.

### 3.4.2 Partitioning uncertainty: parameter, model structure and climate change scenario

Across Brazil for forecasts of live biomass and DOM change to 2100, the largest proportion of uncertainty is derived from parameters (biomass = 0.62, DOM = 0.68), followed by model structure (biomass = 0.21, DOM = 0.24), with the remaining due to climate change scenario uncertainty (biomass = 0.16, DOM = 0.18) (Figure 7). However, there were important spatial variations in uncertainty contributions both between and within biomes. Parameter uncertainty was on average the largest across the Amazon (biomass = 0.74, DOM = 0.72) and Atlantic forest (biomass = 0.71, DOM = 0.66), with smaller contributions in the

Cerrado (biomass = 0.64, DOM = 0.52) and Caatinga (biomass = 0.49, DOM = 0.38; Figure 7). Structural uncertainty follows the inverse spatial pattern, contributing its largest component (though still smaller than parameters) across the Cerrado (biomass = 0.20, DOM = 0.30) and Caatinga (biomass = 0.29, DOM = 0.36). Uncertainty due to climate change scenarios was relatively large in Cerrado and Caatinga (0.11-0.26), but was still the minority contribution in all cases. Moreover, in absolute uncertainty terms, structural (range = 0.4-1.0 MgC ha$^{-1}$) and climate change (range = 0.3-0.7 MgC ha$^{-1}$) uncertainty vary relatively little

compared to parameter uncertainty (range = 0.9-4.9 MgC ha$^{-1}$) which has its largest absolute uncertainties across the Amazon and Atlantic forests with smaller values over the Cerrado and Caatinga (Figure A15). A reduction of parameter uncertainty (~75%) in the Cerrado and Caatinga relative to the Amazon may be linked with these areas being more water-limited (Figure A2), providing a smaller ecologically realistic parameter space (as defined by the EDCs; see SI Text S1.1 for details) and thus reducing uncertainty.





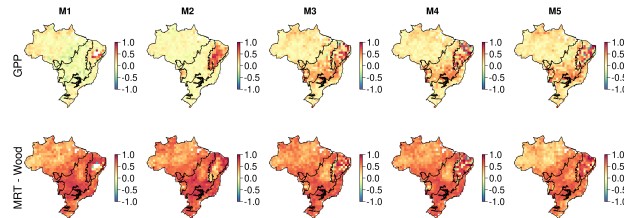

**Figure 8.** Correlation maps between the simulated change in biomass stocks (SSP2-4.5; 2001-2100) and ecosystem variables. These maps identify spatial variation in the sensitivity of biomass change to key ecosystem variables. Correlates are wood mean residence time (MRT; years) and gross primary productivity (GPP; gC m$^{-2}$ day$^{-1}$) estimated across the whole simulation period.

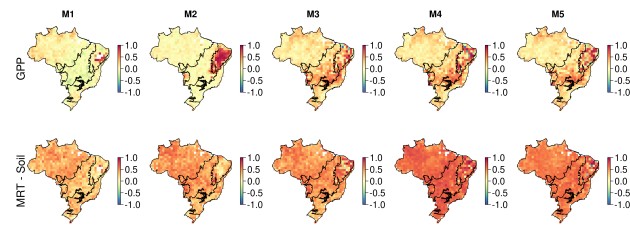

**Figure 9.** Correlation maps between the simulated change in DOM stocks (SSP2-4.5; 2001-2100) and ecosystem variables. These maps identify spatial variation in the sensitivity of biomass change to key ecosystem variables. Correlates are soil mean residence time (MRT; years) and gross primary productivity (GPP; gC m$^{-2}$ day$^{-1}$) estimated across the whole simulation period.

### 3.4.3 Quantifying the determinants of future carbon stock change

Our analysis highlights substantial differences in the magnitude and spatial variation of correlations between ecosystem variables for both biomass (Figure 8, A16) and DOM (Figure 9, A17). Overall, both simulated biomass and DOM change are most strongly correlated with NPP$_{wood}$, MRT$_{wood}$ and MRT$_{soil}$ with only small variations in this pattern among models.

On average MRT$_{wood}$ (r = 0.57) is the most important correlate with biomass change, followed by NPP$_{wood}$ (r = 0.31), MRT$_{soil}$ (r = 0.27; Figure 8) and GPP (r = 0.25). There is substantial spatial variation with NPP$_{wood}$ being the dominant correlate in 12-29% of pixels (between model variation). MRT$_{wood}$ is strongly correlated with biomass change (r > 0.9) across most of Brazil, except the Amazon and isolated areas of the Cerrado and Caatinga where the correlation coefficient declines (r < |0.25|). In contrast, NPP$_{wood}$ is the process most strongly correlated with biomass in Caatinga and parts of the Amazon, consistent for all models. There are some between-model differences indicating varied sensitivity to ecosystem variables (Figure 8). For example, GPP becomes on average an increasingly important correlate as model complexity increases (from M1 = 0.16 to M5 = 0.30). Litter MRT has a low correlation value (M1-4 = ∼0.01) for all models except M5 (which includes wood litter, M5 = 0.06), which has increased correlation in Caatinga (M1-4 = ∼0.03, M5 = 0.18), the Atlantic forest (M1-4 = ∼0.01, M5 = 0.11) and Pampa (M1-4 = ∼0.01, M5 = 0.12).



Uncertainty in DOM stock change is most strongly correlated with $MRT_{soil}$ (r = 0.5), $NPP_{wood}$ (r = 0.36) and $MRT_{wood}$

(r = 0.33; Figure 9). GPP (r = 0.17 to 0.30) and $MRT_{soil}$ (r = 0.36 to 0.56) become increasingly correlated as model complexity increases. $MRT_{wood}$ and $NPP_{wood}$ correlation remains similar between models. There is substantial spatial variation and between-models differences. For example, $NPP_{wood}$ is more strongly correlated with DOM change than $MRT_{soil}$ in 10-44 % of pixels, concentrated over the Cerrado and western Amazon. As with simulated biomass change, $MRT_{litter}$ correlation increases in model M5.

## 390   4   Discussions

### 4.1   Does increasing model complexity improve agreement with evaluation information?

Increasing model complexity by including a wood litter pool (M5) reduces bias (by $\sim$ 65 %) and spatial consistency (M1-4 = $\sim$40 to M5 = $\sim$50 %) between total fire emissions estimated by DALEC and independent estimates increases substantially on the inclusion of a wood litter pool in M5 (Table 2, Figure 5) and increases the spatial consistency from . However, the

DALEC estimated fire emissions uncertainty increased (Table 2) and there remains a persistent spatial disagreement across the southern Cerrado / Amazon boundary (Figure 4). The assimilation of repeat estimate EO biomass should help resolve carbon stock trajectories and provide constraint on the wood litter pool determining the available fuel load for fire combustion (Smallman et al., 2017; Quegan et al., 2019). Fire has a significant impact on the Brazilian C-cycle reducing net carbon uptake by up to 30% (Table 2) but with substantial inter-annual variation (Figure 5). Thus, inclusion of wood litter is potentially a

major step forward in improving DALECs fire emissions given appropriate observational constraint. Wood litter has received comparatively little attention in TEMs, but plays an important role in ecosystem carbon and nutrient cycling (Brovkin et al., 2012; Magnússon et al., 2016).

Contrary to our hypothesis net carbon uptake was not reduced by the inclusion of a water cycle (Table 2). The inclusion of water stress between M2 and M3-5 reduces GPP by up to 30%, so the hypothesised sensitivity of simulated C uptake to soil

moisture was confirmed. However, these reductions in GPP were compensated by increases in C residence times reducing C release from decomposition, therefore the modelled net carbon exchange was not strongly affected (Figure A13).

DALEC and CTE estimate that net carbon uptake increased between 2001 and 2017 (Figure 5) are consistent with the net increase in LAI observed by Copernicus and reproduced by each DALEC model (Figure 3). The observed net increase in LAI is consistent with that found across the globe and has been associated with the accumulation of atmospheric $CO_2$ (Zhu et al., 2016;

Piao et al., 2020). In our analyses, the trend in NEE is concurrent with changes in GPP and respiration over time but dominated by GPP (Table 2; Figure A8). CTE NEE estimates are consistent with other atmospheric inversion and TEM intercomparisons (e.g., Jung et al., 2020; Zscheischler et al., 2017). This consistency among TEM and atmospheric inversion supports robustness of the increasing sink strength estimated by DALEC, but also that the DALEC models are likely overestimating current net uptake across the Amazon (Figure A7).

There is consistent evidence of a net increase in LAI (Zhu et al., 2016; Piao et al., 2020). However, disagreement remains as to whether GPP is also increasing, especially in tropical forests. Whether a given analysis indicates increasing GPP concurrent





with LAI is linked to the sensitivity of the respective models to variations in atmospheric $CO_2$ (Melnikova and Sasai , 2020; Sun et al., 2019). The sensitivity of GPP to $CO_2$ remains poorly quantified, especially in the tropics, due to the lack of in-situ data and large observational uncertainties (Sun et al., 2019). As a result, process-model-enhanced GPP may be overestimated,

due to missing nutrient limitation (He et al., 2017), acclimation processes (Ainsworth & Rogers, 2007) or errors in the coupling of the water and C cycles (Wang et al., 2020). Our analyses estimate increases in both LAI and GPP for much of Brazil, but declines in water-limited regions (e.g. Caatinga) (Figure A8, A9), consistent with other studies using alternate methodologies (He et al., 2017; Jung et al., 2020; Sun et al., 2018; Zhang et al., 2019).

## 4.2 What is the relative importance of parameter, structure and climate change scenario on projections of future
carbon stocks?

The five DALEC models and their parameter ensembles were projected to the year 2100 under four climate change scenarios (Figure 6). The resultant multi-model ensemble of carbon cycle trajectories shows a similar spread and behaviour as that found in regional model intercomparisons (e.g., Arora et al., 2020). For example, by 2100 the ensemble contains models which are still accumulating carbon, have plateaued and those which have peaked and are now declining (Figure 6; Friend et al., 2014;
Arora et al., 2020). Given the similar behaviours we found, meaningful lessons can be drawn from our analysis to inform ESM/TEM model inter-comparisons.

Uncertainty associated with projections of biomass and DOM dynamics is dominated by parameter uncertainty, followed by model structure and climate change scenario. For parameter uncertainty there are also clear spatial patterns, particularly focused over the Amazon and Atlantic forests (Figure 7, S15), while spatial patterning is weaker for model structure and
climate change scenario. The uncertainty associated with parameters alone precludes determination of whether Brazil is a net sink or source of carbon at the 90 % confidence level (Figure 6). In order to reduce projection uncertainties there has been substantial effort within the ESM and TEM communities on identifying the underlying processes (i.e. structure) driving model error at a range of scales (e.g., Friend et al., 2014; De Kauwe et al., 2014; Zhou et al., 2018) and methodologies to weight existing analyses based on their simulation of contemporary observation constraints (e.g., Wenzel et al., 2014; Exbrayat et al.,
2018a). However, increasing model complexity also increases the number of parameters that may be poorly constrained in the absence of adequate observational data (Prentice et al., 2015). Our framework for exploring the relative impact of parameter uncertainties on future carbon dynamics contrasts strongly with the typical process of model development. Typically in TEM development relatively small model structural changes are made and the resultant response being extensively investigated in the absence of knowledge on parameter uncertainties (e.g. Mercado et al., 2009; Verheijen et al., 2015; Jones et al., 2020).
Uncertainties in terrestrial carbon cycling have remained large over multiple inter-comparison cycles (Arora et al., 2020). Thus, we argue that a greater focus is needed on refining parameters themselves.

## 4.3 Which ecosystem traits are most strongly correlated with simulated carbon dynamics?

$MRT_{wood}$ was the parameter most tightly coupled to the response of biomass C stocks to climate change between now and 2100 (Figure 8), and the third most important determinant of the response of DOM (Figure 9). This high importance is likely a



function of three factors. First, there is a relatively weak constraint on the $\text{MRT}_{wood}$ parameter in each pixel over the calibration period due to lack of repeat observations of wood biomass. Second, woody biomass is typically the largest biomass pool, and $\text{MRT}_{wood}$ is a key control on turnover and therefore on decadal changes in the size of this pool. Third, $\text{MRT}_{wood}$ is a key control on C inputs to the soil C and wood litter (M5 only) pools in DALEC, generated by modelled wood losses. Similarly it is important that $\text{NPP}_{wood}$ is the second most important determinant of future dynamics of both biomass and

DOM. Efforts to constrain estimates of $\text{MRT}_{wood}$ and NPP allocation are thus critical for more robust predictions of C storage (Friend et al., 2014; Koven et al., 2015; Zhou et al., 2018). These efforts should be enhanced by current and future missions that repeatedly measure woody biomass from space (Quegan et al., 2019), using approaches like CARDAMOM for model calibration (Smallman et al., 2017). Robust assessments of wood biomass uncertainty in these EO products will be critical to producing more constrained C cycle projections (Exbrayat et al., 2019).

Higher resolution studies (e.g. at ha or $\text{km}^2$ resolution) over areas of rapid biomass change, such as the arc of deforestation in Brazil, will provide added insights into model structure and parameter uncertainties. There are challenges for the CARDAMON-DALEC approach to work at finer scales and with more dynamic ecosystems. For instance, the concept of MRT is less appropriate in modelling successional change and highly dynamic systems, where internal feedbacks may adjust C losses (mortality) with variations in density and age (Peters , 2003; Ge et al., 2019). Chronosequence data at high spatial and

temporal resolutions can provide the means to test alternate representations of successional variation in C cycling and storage (e.g., Safar et al., 2020).

Variations in C storage linked to model structure were smaller than those linked to model parameterisation, except in specific areas of Brazil (Caatinga; Figure 7, S15). The selection of five model structures was limited by our choice, so it is perhaps not surprising that the parameter calibration, which allows for multidimensional variation over broad priors, contributes more

variation to projections than does model structural variability. However, the variation in model structure was designed to test whether hypothesised key processes were important in projections. For instance, we used models with and without a water cycle simulation to test the importance of carbon-water feedbacks in projections of C storage to 2100. Models M3-5 included dynamic simulation of soil moisture changes and its interactions with canopy processes. Projections with these models thus included the potential development of soil moisture stress, with an impact on GPP. Models M1 and M2 had no direct effect

of soil moisture on C cycling. This soil moisture feedback on GPP only manifested itself in projections for north east Brazil, the driest part of the country, in the Caatinga biome, and some nearby parts of Cerrado (Figures 4, S7). This feedback does have an impact on projected C storage (Figure 7; Table A2), but these effects are of similar or less magnitude to parameter uncertainty. We conclude that for much of Brazil, outside of Caatinga, our model-data fusion shows a limited sensitivity of C cycling to future soil moisture stress. This is likely a result of $CO_2$ fertilisation leading to reductions in plant water demand

that are explicit in both ACM GPP models. However, it is possible that land surface models like DALEC are overestimating $CO_2$ fertilisation effect (Wang et al., 2020) highlighting the need for further evaluation and refinement.

As expected climate change scenario uncertainty is dwarfed by uncertainties in both model structure and parameters (Figure 7). However, we have only used projections from one earth system model (UKESMv1), and therefore we have not quantified the impact of uncertainty in meteorology / climate change itself for a given emissions scenario. While uncertainty in meteorology





has been shown on longer time scales (decadal) to contribute a minority component relative to the overall uncertainty (e.g., Huntingford et al., 2013; Bonan et al., 2019), we do consider multi-model forcing well worth including in future analyses to provide the most robust assessment of observation constrained carbon trajectories possible.

## 4.4    Future avenues to improve observational constraint

Improving constraints on NPP and MRT will reduce uncertainties associated with simulated biomass and DOM change (Figure
8-9). The reverse is also true: providing information on contemporary biomass and DOM dynamics will improve constraint on NPP and MRT, which in turn reduces uncertainty when simulating into the future. In addition to repeat measurement of biomass stocks there are a diverse range of alternate observational constraints which could provide critical information.

    Assimilation of atmospheric inversion estimated net carbon exchange could provide constraint on simulated biomass and DOM change and therefore on their correlated ecosystem variables and parameters. The divergence between estimates of net
carbon exchange by different atmospheric inversion systems has reduced substantially over recent years (Gaubert et al., 2019). Moreover, given sufficient observational constraint, posterior estimates of net exchange can converge even with substantially different priors of biospheric exchange indicating a robust analysis (White et al., 2019). But this approach limits ecological learning (e.g. refining MRT to reduce prediction uncertainties). Direct assimilation of atmospheric observations of $CO_2$ concentrations into TEMs has previously been used to refine a limited number of parameters for specific plant functional types
(PFT) (e.g., Reuter et al., 2011). However, key ecosystem traits (e.g. NPP allocation and MRTs) vary within what would classically be considered the same PFT (e.g., Exbrayat et al., 2018b) necessitating the development of strategies to allow direct model parameterisation at sub-PFT scales. We recognise that this approach will have substantial technical and computational challenges but the potential benefits are too great to ignore.

    Methodologies to estimate potential biomass, i.e. in the absence of direct human disturbance (Exbrayat and Williams ,
2015), or potential regrowth rates (Cook-Patton et al., , 2020) have recently gained attention. However, a key weakness of these estimates is their dependence on current biomass $\sim$ climate relationships, thus lacking the ability to project into new climate or disturbance regimes (Lewis et al., 2019). Nevertheless, we see an opportunity to use potential biomass information as an additional constraint, in conjunction with existing EO biomass maps, on the steady state generated by a given combination of current climate and parameters in the absence of human disturbance (which can be determined analytically). As simulation
of biomass change correlates strongly with both $MRT_{wood}$ and $NPP_{wood}$ (Figure 8), we expect that assimilation of potential biomass will also provide constraints on these parameters, increasing confidence in simulations of climate-sensitive carbon cycle trajectories.

    Forest biomass removal has a significant impact on the Brazilian C-cycle, resulting in losses between 100-450 TgC yr$^{-1}$ (Table 2; Figure A10) and the subsequent regrowth of secondary forests. Secondary forests across the Brazilian Amazon alone
are estimated to cover an area of 22-28 Mha accumulating 1.5-11.25 MgC ha yr$^{-1}$ but are estimated to be re-cleared every 5-10 years (Poorter et al., 2016; Yang et al., 2020). It is likely that we are missing losses driven by degradation, re-clearance events, and edge effects (e.g., Yang et al., 2020)(e.g. Yang et al., 2020) that are not accounted for in existing EO datasets, such as GFW, that are used to drive disturbance in our models (Milodowski et al., 2017; Silva Junior et al., 2020). Missing



these disturbance events would lead to overestimation of long term accumulation of woody carbon, consistent with the likely
overestimate of net carbon uptake estimated by the DALEC models already discussed in comparison with CTE. Moreover,
improved disturbance drivers can add additional constraint to the C-cycle, potentially reducing uncertainty and refining our
best estimates of key ecosystem traits as has previously been demonstrated due to the inclusion of fire disturbance information
(Exbrayat et al., 2018b). Therefore, improvements in EO-based estimates of deforestation, degradation and the inclusion of
re-clearance information is essential to reflect their associated emissions and therefore improve model calibration efforts.

## 525    5    Conclusions

We use a MDF approach in conjunction with 5 related terrestrial carbon cycle models, with differences in key feedbacks and
processes, and observational constraints to quantify the current and future state, trajectory and uncertainties of the Brazilian
carbon cycle. Our analysis shows that parameter uncertainty exceeded both the structural uncertainty captured within our
model ensemble and uncertainties in projected climate except in drier areas of the country. Parameter uncertainty alone was
large enough to span the source / sink boundary identifying a clear need to further refine parameter constraint not just model
structural complexity. We identify $NPP_{wood}$, $MRT_{wood}$ and $MRT_{soil}$ as key uncertainties influencing future trajectories. Given
the bi-directional nature of these associations we have identified future avenues for new observational constraints on these
ecosystem properties. Such constraints include repeat estimations of AGB, estimates of NEE from atmospheric inversion,
estimates of potential AGB stocks and improved estimates of fluxes driven by ecosystem disturbance and regrowth. Improving
constraints on residence times will greatly improve our ability to make meaningful predictions into the future.

*Code and data availability.*    The model outputs parameters and carbon cycle outputs are freely available to download from doi:https://doi.org/10.7488/ds/3000.
CARDAMOM and DALEC source codes are available to download from a Github repository https://github.com/GCEL/CARDAMOM with
registration provided on request to either T. L. Smallman or M. Williams.

## Appendix A:  Further methods

### 540    A1    Description of the CARDAMOM framework

CARDAMOM is a MDF framework which uses a Bayesian approach to estimate ensembles of DALEC model parameters
(x; Table A1) consistent with observational constraints and their uncertainties. The likelihood (i.e. probability) of a given x
is estimated with respect to the assimilated observations (P(x|O)) as a function of the likelihood of the observations given the
current parameters (P(O|x)) and any prior knowledge. In our analyses we assume a prior range on each parameter defined as
$P_{range}$(x) and ecological and dynamical constraints (EDCs) estimated as a function of DALEC output (PEDC(DALEC(x)))
(See   Bloom and Williams , 2015,  for details). Finally, all models apply a prior value ($P_{prior}$(x)) on the ratio of $R_a$ to





GPP ($R_a$:GPP) of 0.46±0.12 (Collalti and Prentice , 2019). The $R_a$:GPP is also known as the carbon use efficiency (CUE = NPP/GPP, $R_a$:GPP = 1-CUE).

$$P(x|O) \propto P(O|x) \cdot P_{range}(x) \cdot P_{prior}(x) \cdot PEDC(DALEC(x)) \tag{A1}$$

$P(O|x)$ is calculated by comparing the $n^{th}$ observation ($O_n$) to the corresponding model state variables or flux ($M_n$) and scaled by the observation uncertainty ($\sigma_n$).

$$P(O|x) = exp(-0.5 \cdot \epsilon (O_n - M_n)^2 / \sigma_n^2) \tag{A2}$$

     CARDAMOM uses EDCs to ensure ecological realism in the accepted parameter sets that are challenging to constrain directly with a numerical prior (Bloom et al., 2016). The EDCs include preventing inappropriate parameter combinations e.g.
fine root residence times being longer than wood. PEDC(DALEC(x)) = 1 when all EDC conditions are achieved otherwise PEDC(DALEC(x)) = 0 and the proposed x is rejected.

### A2   Summary description of DALEC models

The structure of each DALEC model follows a similar system of carbon stocks and fluxes (Figure 2). Carbon enters the system via GPP which is estimated as a function of leaf area (as simulated by DALEC), canopy photosynthetic efficiency
(estimated per pixel), absorbed solar radiation, atmospheric $CO_2$ concentration and air temperature. GPP is allocated to Ra and live tissues based on fixed fractions. Canopy growth is determined by a combination of direct allocation from GPP and release of carbon from the labile pool, which is controlled by a day of year model. Similarly, canopy senescence is determined by a day year model with a parameterised leaf life-span. Wood and fine roots follow continuous turnover based on first order kinetics. Decomposition of dead organic matter and associated $R_{het}$ follows first order kinetics with an exponential temperature
sensitivity. Using EO derived information on burned area fire emissions (FIRE) are based on available carbon stocks. From these, key emergent fluxes are derived including net primary productivity (NPP = GPP - $R_a$), net ecosystem exchange of $CO_2$ (NEE = $R_a$ + $R_{het}$ - GPP) and net biome exchange (NBE = NEE + FIRE). Model M1 is DALEC as described in Bloom and Williams (2015). Each subsequent model cumulatively increases the complexity of its process representation in the following ways.

In M2 the photosynthesis model (ACM1; Williams et al., 1997) is replaced with a revised version (ACM2; Smallman and Williams , 2019). The key new feature of ACM2 is a stomatal conductance model which explicitly balances supply of water via the roots and atmospheric demand, estimated as a function of absorbed radiation and vapour pressure deficit (VPD). ACM2 also includes a full water cycle simulating evaporation, drainage and runoff. However to isolate the impact of the stomatal conductance model itself the soil moisture is fixed at saturation. In M3 soil moisture is dynamic allowing explicit simulation
of the water cycle including the potential for development of drought.

     In M4, rather than estimating $R_a$ as a fixed fraction of GPP ($R_a$:GPP), $R_a$ is divided between that associated with maintenance ($R_m$) or growth ($R_g$). $R_m$ is estimated as a fixed fraction of GPP while $R_g$ is estimated as 22% of carbon allocated to live





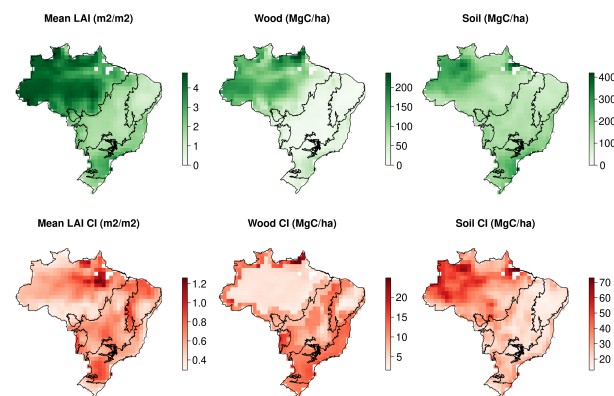

**Figure A1.** Maps of observational constraints and their associated uncertainties used in the CARDAMOM assimilation framework. Mean annual LAI, wood stocks and initial soil stocks.

tissue (equivalent to 28% of NPP; Waring and Schlesinger , 1985). This development provides the first step in explicit simulation of respiratory costs needed for implementation of economic theory within DALEC (e.g. Thomas et al., 2019; Flack-Prain et al., 2020).

M5 includes a wood litter pool rather than allocating wood litter directly to the soil. The inclusion of a wood litter pool tests our ability to constrain this potentially large but challenging to observe carbon store (Magnússon et al., 2016). Furthermore, wood litter can play an important role in carbon emissions due to fire (vanderwerf et al., 2006) and its inclusion here allows us to investigate whether M5 has an improved estimate of fire emissions.

## Appendix B: Further results

DALEC estimated carbon losses due to forest biomass removals are estimated to vary between 120 - 400 TgC yr$^{-1}$ reducing the biospheric sink by 5 - 32% (Figure 5). The largest biomass extractions are estimated for 2016 (387 - 425 TgC yr$^{-1}$; between model range) and 2017 (302-337 TgC yr$^{-1}$; between model range). In all other years the mean biomass loss was substantially lower at 102-220 TgC yr$^{-1}$. The interannual variation follows the GFW estimate as the fraction forest cover loss is derived by GFW. GFW estimates forest losses are larger than the DALEC models at the beginning of the analysis but converging by 2017 potentially driven by the accumulation of wood in the DALEC models. As GFW is used as a forest loss driver by the DALEC suite this comparison is not fully independent. However, disagreement between these estimates highlights the importance of the biomass map underpinning the analyses (See discussion for further details).



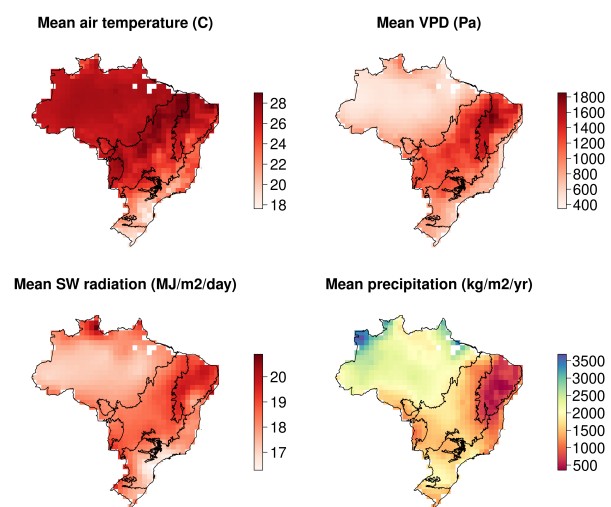

**Figure A2.** Mean meteorological conditions for the calibration period from CRU-JRA dataset (2001-2017).



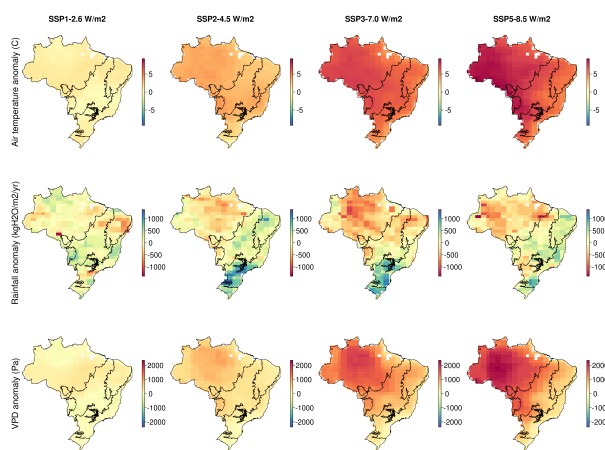

**Figure A3.** Maps of mean climate anomaly averaged between two 5 year periods at the beginning (2018-2022) and end (2095-2099) of the future simulations.

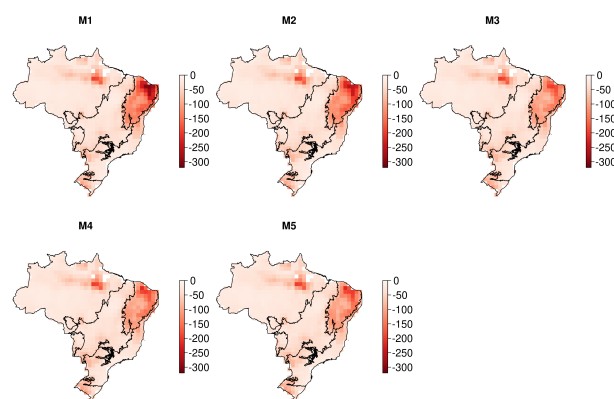

**Figure A4.** Maps showing the pixel level median log-likelihood scores achieved by each. More negative values indicate a greater disagreement between the model simulated carbon cycle and corresponding observational constraints. The Brazil wide averaged likelihood scores are -48.7, -47.7, -45.1, -45.6, -45.7 for models M1-5 respectively.





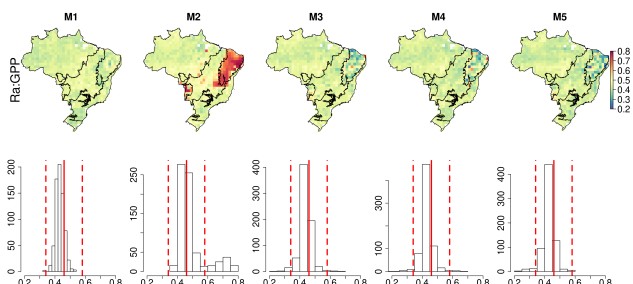

**Figure A5.** Comparison between Ra:GPP estimated by DALEC models and prior value assimilated by CARDAMOM. The top row shows the maps of the pixel level median estimated by each model. The bottom row shows the histograms of the mapped information. The vertical solid red line shows the prior value and the vertical dashed red line shows the uncertainty range associated with the prior value.



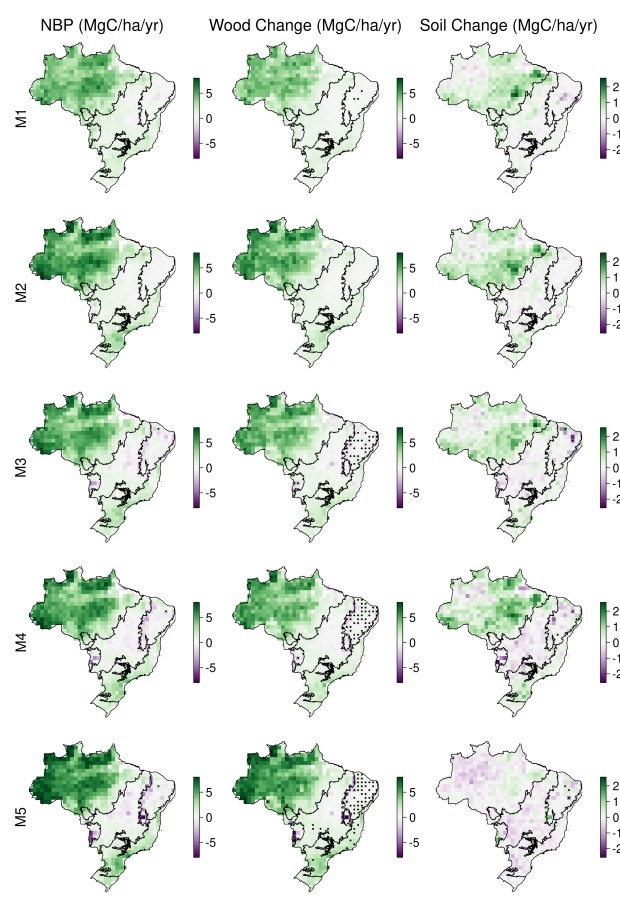

**Figure A6.** DALEC suite estimates of NBP (2001-2017) (i.e. -NBE to provide sign consistency with C stocks), wood and soil carbon stock change. Stippling to indicate >90% confidence on a given pixel being a net source or sink of carbon during our analysis.

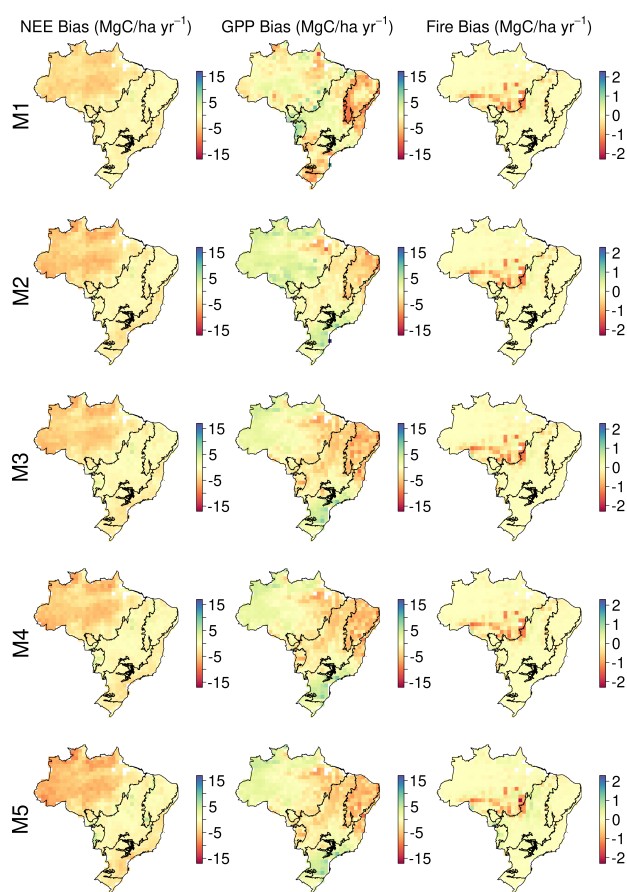

**Figure A7.** Maps showing the mean bias between each DALEC model and independent data for NEE, GPP and fire emissions. See Figure 4 for details.



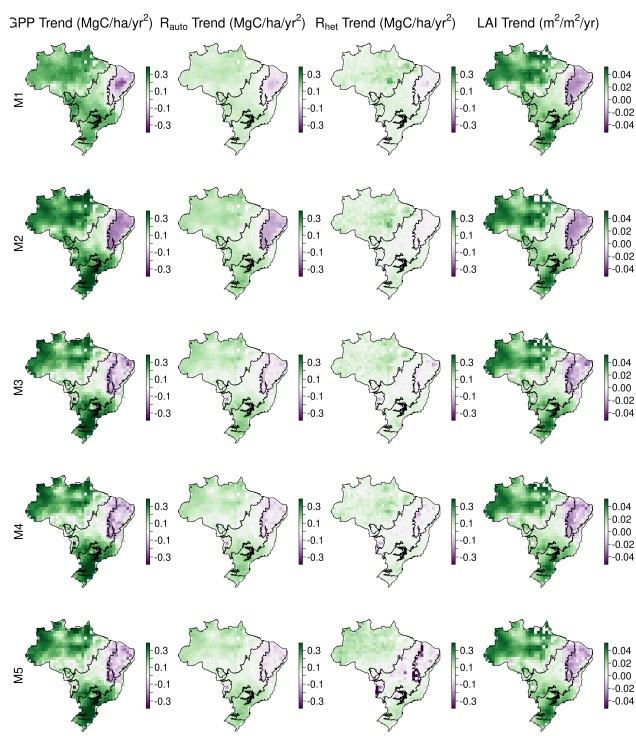

**Figure A8.** Model specific trends (2001-2017) in GPP, $R_{auto}$, $R_{het}$, LAI estimated from the pixel level median estimates.



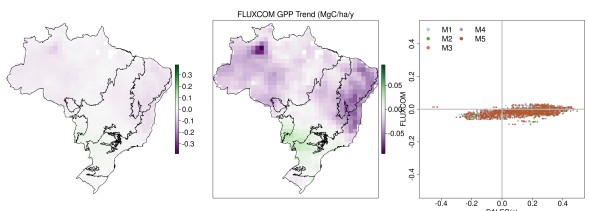

**Figure A9.** FLUXCOM GPP trend across Brazil (2001-2017). The map on the left shows the FLUXCOM trend on the same axis as the DALEC models in Figure A8. The centre map shows the same information but on an axis based on FLUXCOM estimates only to highlight the spatial pattern. The panel on the right shows a comparison between trends estimates by FLUXCOM and the DALEC models ($R^2$ = 0.18-0.21).



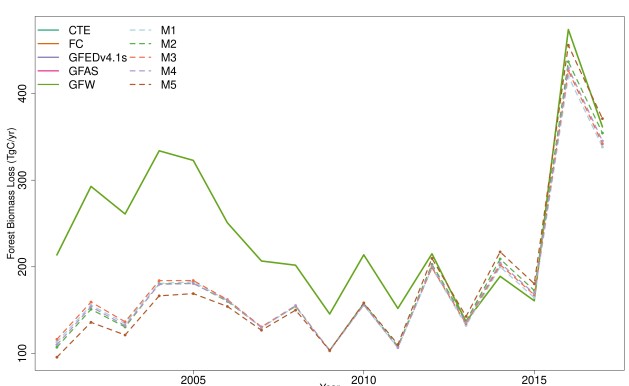

**Figure A10.** Comparison of DALEC estimated forest biomass loss.





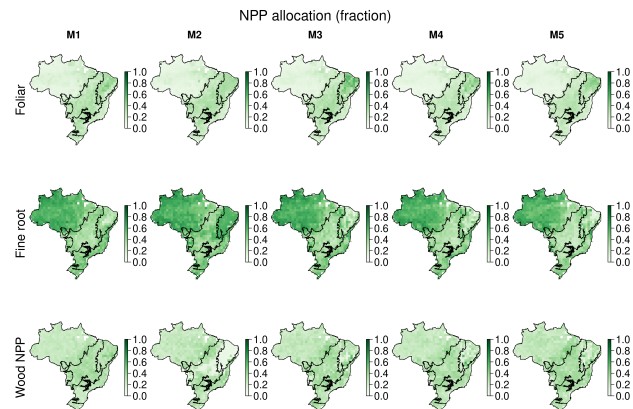

**Figure A11.** Maps of the localised median estimates of NPP allocation fractions to live tissues for each model.



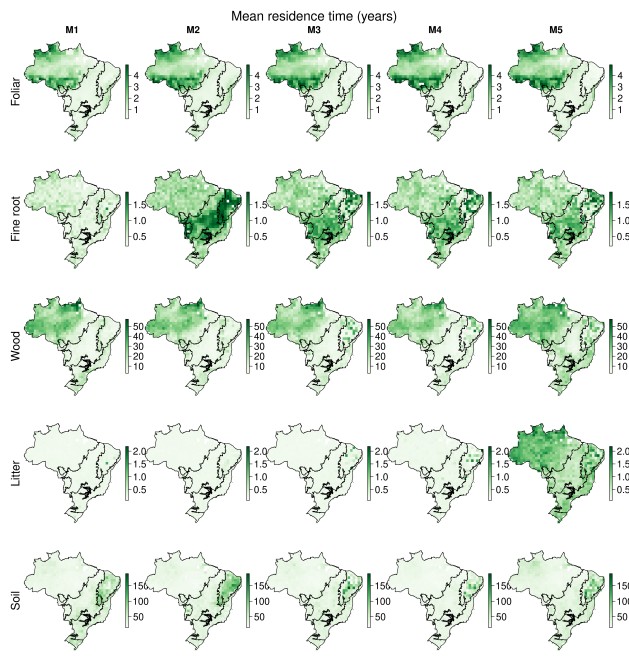

**Figure A12.** Maps showing the localised median estimates of carbon pool mean residence times (MRT) for each model. Note that for models M1-4 litter MRT is foliar and fine root litter, while M5 also includes wood litter.

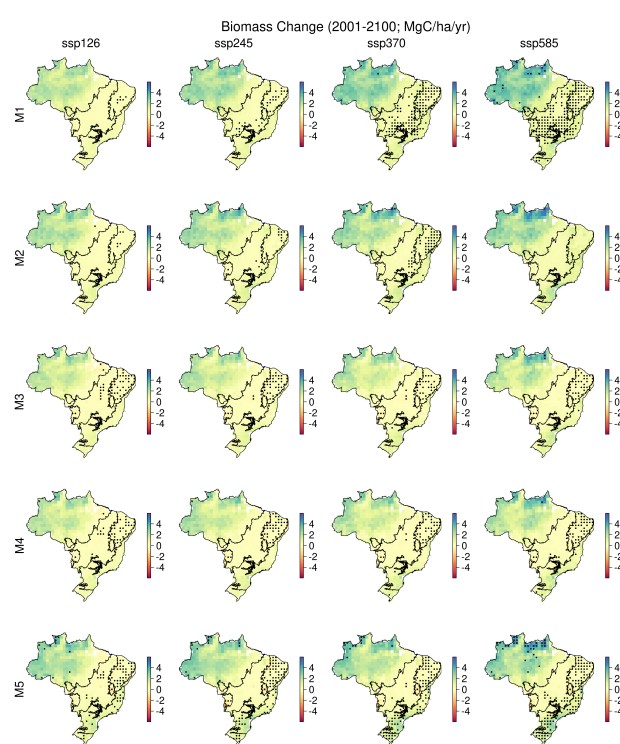

**Figure A13.** Maps of DALEC model estimates biomass (sum of labile, foliage, fine root and wood) change between 2001 and 2100. The mapped values are estimated from the pixel level median estimates. Areas with biomass change with >90% confidence are shown with stippling.

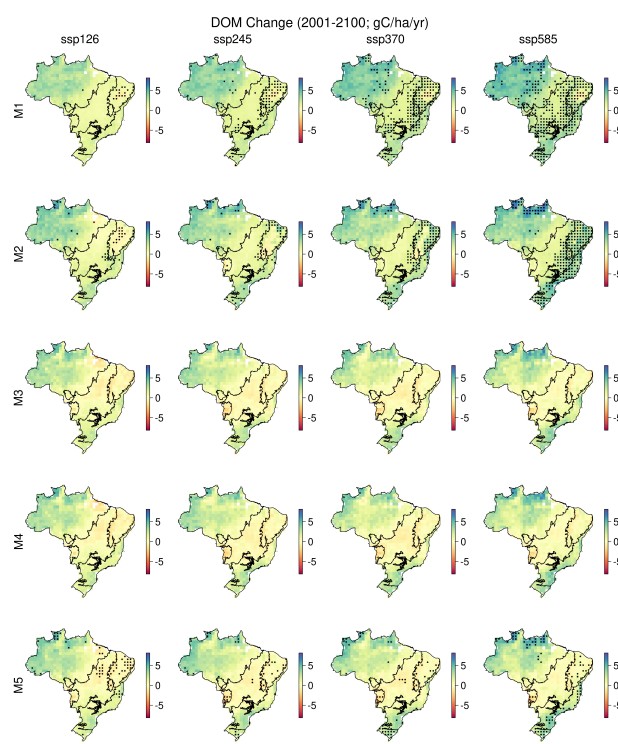

**Figure A14.** Maps of DALEC model estimates DOM (sum of soil, litter and in M5 wood litter) change between 2001 and 2100. The mapped values are estimated from the pixel level median estimates. Areas with DOM change with >90% confidence are shown with stippling.



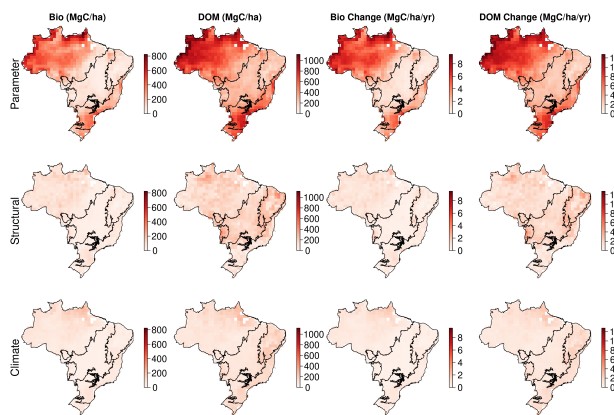

**Figure A15.** Absolute uncertainty range (5 - 95% quantiles) simulated for each pixel in 2100 attributed to model parameters, model structural diversity and climate change scenario.

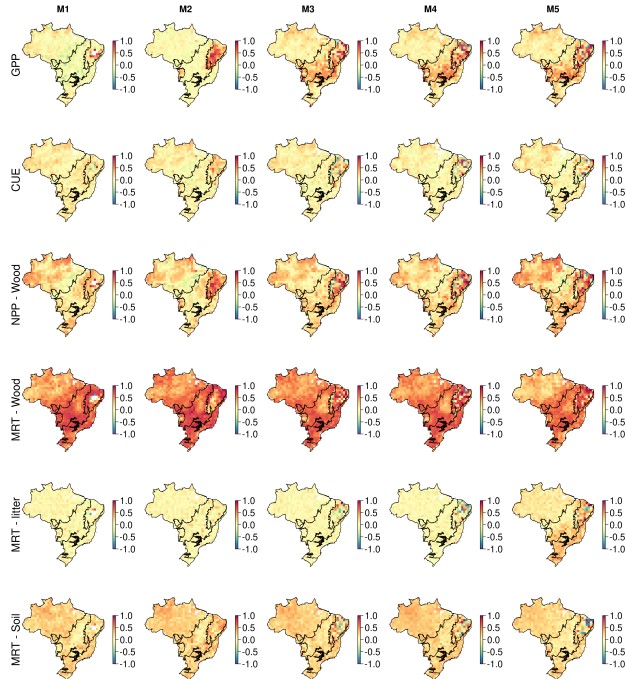

**Figure A16.** Correlation maps between the simulated change in biomass stocks (SSP2-4.5; 2001-2100) and ecosystem variables. These maps identify spatial variation in the sensitivity of biomass change to key ecosystem variables. Correlates are pool specific mean residence time (MRT; years), net primary productivity (NPP; gC m$^{-2}$ day$^{-1}$), carbon use efficiency (CUE = 1-Ra:GPP) and gross primary productivity (GPP; gC m$^{-2}$ day$^{-1}$) estimated across the whole simulation period.





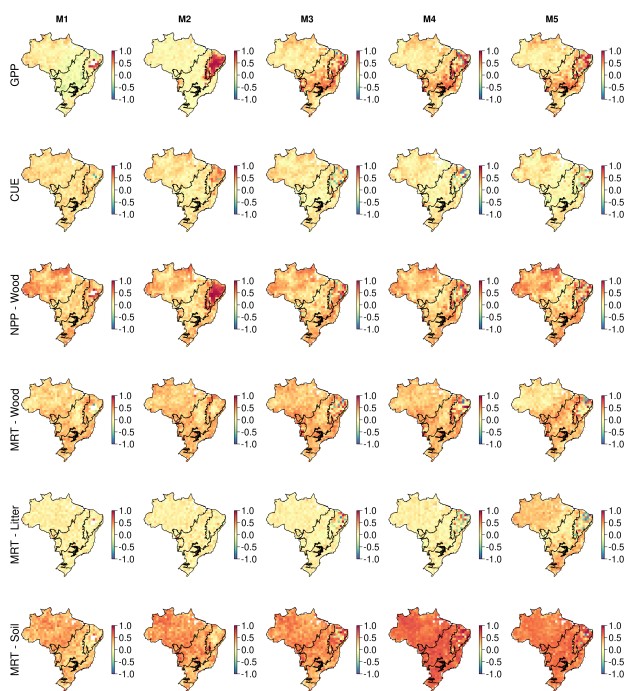

**Figure A17.** Correlation maps between the simulated change in DOM stocks (SSP2-4.5; 2001-2100) and ecosystem variables. These maps identify spatial variation in the sensitivity of biomass change to key ecosystem variables. Correlates are pool specific mean residence time (MRT; years), net primary productivity (NPP; gC m$^{-2}$ day$^{-1}$), carbon use efficiency (CUE = 1-Ra:GPP) and gross primary productivity (GPP; gC m$^{-2}$ day$^{-1}$) estimated across the whole simulation period.



**Table A1.** Description of parameters estimated for each DALEC model, each parameter is given a name, unit, description. As not all parameters are used by all models the applicable models are also given following the code used in the main text. Note: Gross Primary Productivity = GPP, Autotrophic respiration = $R_a$, Autotrophic maintenance respiration = $R_m$, heterotrophic respiration = Rh. Litter is assumed to be the combined foliage and fine root litter pools, where appropriate wood litter will be explicitly stated. Note that GPP allocation fractions are applied sequentially such that GPP allocation to $C_{wood}$ = GPP - (GPP·$R_a$:GPP) - (GPP·$GPP_{lab}$) - (GPP·$GPP_{root}$).

| Name | units | Description | Model(s) |
|---|---|---|---|
| $R_a$:GPP | fraction | Fraction of GPP allocated to $R_a$ | M1-3 |
| $R_m$:GPP | fraction | Fraction of GPP allocated to $R_m$ | M4-5 |
| $GPP_{fol}$ | fraction | Fraction of GPP allocated to foliage | M1-5 |
| $GPP_{lab}$ | fraction | Fraction of GPP allocated to labile | M1-5 |
| $GPP_{root}$ | fraction | Fraction of GPP allocated to fine root | M1-5 |
| Leaf lifespan | years | Maximum natural leaf lifespan | M1-5 |
| Leaf growth day | day of year | Julian day on which max labile turnover to foliage as defined by the phenology model | M1-5 |
| Leaf growth period | days | Standard deviation defining the period over which labile turnover to foliage occurs | M1-5 |
| Leaf fall day | day of year | Julian day on which max foliar turnover to litter as defined by the phenology model | M1-5 |
| Leaf fall period | days | Standard deviation defining the period over which foliar turnover to litter occurs | M1-5 |
| Wood turnover | $day^{-1}$ | Fraction of wood loss per day | M1-5 |
| Fine root turnover | $day^{-1}$ | Fraction of fine root loss per day | M1-5 |
| Litter decomposition | $day^{-1}$ at $0^oC$ | Fraction of fine root loss per day | M1-4 |
| Decomp efficiency | fraction | Fraction of litter and wood litter turnover allocated to soil | M5 |
| Litter mineralisation | $day^{-1}$ at $0^oC$ | Baseline litter turnover to $R_{het}$ | M1-4 |
| Litter turnover | $day^{-1}$ at $0^oC$ | Baseline litter turnover applied in conjunction with "Decomp efficiency" | M5 |
| Wood Litter turnover | $day^{-1}$ at $0^oC$ | Baseline wood litter turnover applied in conjunction with "Decomp efficiency" | M5 |
| Soil mineralisation | $day^{-1}$ at $0^oC$ | Baseline soil turnover to $R_{het}$ | M1-5 |
| $R_{het}$ coefficient | - | Exponential temperature response coefficient for $R_{het}$ | M1-5 |
| LMA | $gCm^{-2}$ | Leaf mass per unit leaf area | M1-5 |
| Ceff | $gCm^{-2}day^{-1}$ | Potential photosynthetic activity per unit leaf area | M1-5 |
| Coarse root fraction | fraction | Fraction of wood assumed to be coarse root. Used in determining rooting depth | M2-5 |
| Root depth coefficient | $gm^{-2}$ | Total coarse and fine root biomass required to reach 50% of max rooting depth | M2-5 |
| Max rooting depth | m | Max rooting depth | M2-5 |
| Initial soil water | fraction | Initial soil water content as fraction of field capacity | M3-5 |
| Initial labile | $gCm^{-2}$ | Size of the labile C pool at time step 1 | M1-5 |
| Initial foliage | $gCm^{-2}$ | Size of the foliar C pool at time step 1 | M1-5 |
| Initial fine root | $gCm^{-2}$ | Size of the fine root C pool at time step 1 | M1-5 |
| Initial wood | $gCm^{-2}$ | Size of the wood C pool at time step 1 | M1-5 |
| Initial litter | $gCm^{-2}$ | Size of the litter C pool at time step 1 | M1-5 |
| Initial soil | $gCm^{-2}$ | Size of the soil C pool at time step 1 | M1-5 |
| Initial wood litter | $gCm^{-2}$ | Size of the wood litter C pool at time step 1 | M1-5 |





**Table A2.** Climate change scenario specific estimates of Biomass change (2001-2100) by DALEC for each of the Brazil biomes (Figure 1). Units are PgC and values in parenthesis are the 5 and 95 % quantiles defining the 90 % confidence interval. For clarity values are rounded to the nearest PgC or two significant figures. SSP126 = SSP1-2.6W m$^{-2}$, SSP245 = SSP2-4.5W m$^{-2}$, SSP370 = SSP3-7.0W m$^{-2}$ and SSP585 = SSP5-8.5W m$^{-2}$.

| Model | Scenario | Amazon | Atlantic Forest | Cerrado | Caatinga | Pantanal | Pampa |
|---|---|---|---|---|---|---|---|
| M1 | SSP126 | 63 (-45 / 162) | 4.4 (-5.3 / 19) | 1.5 (-1.3 / 7.6) | 4.5 (-5.3 / 22) | 0.5 (-0.6 / 2.1) | 0.7 (-0.6 / 3.3) |
| | SSP245 | 80 (-35 / 191) | 7.4 (-3.2 / 26) | 2.7 (-0.6 / 11) | 6.9 (-4.0 / 28) | 0.8 (-0.4 / 2.8) | 0.8 (-0.5 / 3.6) |
| | SSP370 | 96 (-26 / 216) | 10 (-1.3 / 33) | 4.0 (0.1 / 14) | 9.1 (-2.8 / 34) | 1.0 (-0.2 / 3.3) | 1.3 (-0.3 / 4.6) |
| | SSP585 | 107 (-20 / 233) | 12 (-0.1 / 38) | 4.6 (0.3 / 16) | 11 (-1.8 / 39) | 1.2 (-0.1 / 4.0) | 1.6 (-0.1 / 5.6) |
| M2 | SSP126 | 66 (-42 / 160) | 4.2 (-5.7 / 19) | 0.73 (-1.8 / 7.1) | 6.0 (-4.4 / 26) | 0.43 (-0.7 / 1.8) | 0.8 (-0.6 / 4.0) |
| | SSP245 | 73 (-38 / 170) | 6.2 (-4.3 / 23) | 2.5 (-1.1 / 13) | 7.9 (-3.3 / 30) | 0.04 (-0.9 / 1) | 1.0 (-0.5 / 4.4) |
| | SSP370 | 79 (-36 / 182) | 7.6 (-3.5 / 27) | 4.4 (-0.1 / 18) | 9.6 (-2.4 / 34) | 0.4 (-0.7 / 1.6) | 1.3 (-0.3 / 5.0) |
| | SSP585 | 88 (-31 / 197) | 9.7 (-2.1 / 32) | 6.7 (-1.2 / 23) | 12 (-1.3 / 41) | 0.4 (-0.7 / 1.7) | 1.6 (-0.2 / 5.9) |
| M3 | SSP126 | 50 (-53 / 140) | 1.3 (-8.6 / 12) | -0.5 (-2.7 / 1.5) | 4.1 (-6.1 / 22) | -0.05 (-1.0 / 1.0) | 0.6 (-0.7 / 3.2) |
| | SSP245 | 54 (-50 / 146) | 1.3 (-8.6 / 12) | -0.1 (-2.4 / 2.4) | 5.1 (-5.6 / 23) | -0.33 (-1.2 / 0.3) | 0.8 (-0.6 / 3.6) |
| | SSP370 | 54 (-51 / 149) | 1.5 (-8.5 / 12) | -0.04 (-2.4 / 2.6) | 6.4 (-4.8 / 26) | -0.3 (-1.2 / 0.4) | 1.2 (-0.4 / 4.5) |
| | SSP585 | 63 (-46 / 164) | 2.9 (-7.3 / 16) | 0.6 (-1.9 / 3.9) | 7.9 (-4.1 / 31) | -0.23 (-1.2 / 0.6) | 1.3 (-0.3 / 4.9) |
| M4 | SSP126 | 52 (-52 / 142) | 1.3 (-8.7 / 13) | -0.46 (-2.5 / 1.9) | 4.5 (-6.0 / 23) | -0.13 (-1.1 / 1.1) | 0.68 (-0.8 / 3.4) |
| | SSP245 | 56 (-49 / 148) | 1.3 (-8.8 / 12) | -0.04 (-2.3 / 2.7) | 5.6 (-5.5 / 24) | -0.37 (-1.3 / 0.4) | 0.9 (-0.7 / 3.8) |
| | SSP370 | 56 (-50 / 151) | 1.5 (-8.6 / 13) | 0.08 (-2.2 / 3.0) | 6.8 (-4.7 / 27) | -0.35 (-1.2 / 0.6) | 1.3 (-0.5 / 4.7) |
| | SSP585 | 65 (-45 / 166) | 2.9 (-7.4 / 16) | 0.69 (-1.7 / 4.3) | 8.4 (-4.0 / 32) | -0.3 (-1.2 / 0.8) | 1.4 (-0.4 / 5.1) |
| M5 | SSP126 | 71 (-29 / 150) | 3.5 (-11 / 15) | -0.29 (-3.2 / 2.3) | 8.8 (-3.0 / 27) | -0.20 (-2.0 / 1.0) | 1.35 (-0.2 / 4.1) |
| | SSP245 | 75 (-26 / 157) | 3.3 (-11 / 14) | 0.21 (-2.9 / 3.4) | 10 (-2.4 / 29) | -0.53 (-2.2 / 0.3) | 1.56 (-0.02 / 4.6) |
| | SSP370 | 75 (-27 / 160) | 3.6 (-11 / 15) | 0.29 (-2.9 / 3.6) | 12 (-1.6 / 32) | -0.49 (-2.2 / 0.4) | 2.1 (0.2 / 5.6) |
| | SSP585 | 86 (-22 / 176) | 5.4 (-9.4 / 19) | 1.1 (-2.4 / 5.2) | 14 (-0.7 / 38) | -0.4 (-2.1 / 0.6) | 2.27 (0.3 / 6.0) |



**Table A3.** Climate change scenario specific estimates of DOM change (2001-2100) by DALEC for each of the Brazil biomes (Figure 1). Units are PgC and values in parenthesis are the 5 and 95 % quantiles defining the 90 % confidence interval. For clarity values are rounded to the nearest PgC or two significant figures. SSP126 = SSP1-2.6W m$^{-2}$, SSP245 = SSP2-4.5W m$^{-2}$, SSP370 = SSP3-7.0W m$^{-2}$ and SSP585 = SSP5-8.5W m$^{-2}$.

| Model | Scenario | Amazon | Atlantic Forest | Cerrado | Caatinga | Pantanal | Pampa |
|---|---|---|---|---|---|---|---|
| M1 | SSP126 | 110 (-51 / 282) | 20 (-10 / 58) | 6.7 (-3.8 / 20) | 16 (-10 / 49) | 1.8 (-1.2 / 5.5) | 2.8 (-1 / 7.4) |
| | SSP245 | 134 (-37 / 319) | 30 (-5.0 / 75) | 11 (-1.5 / 28) | 22 (-6.5 / 59) | 2.5 (-0.8 / 6.9) | 3.2 (-0.7 / 7.9) |
| | SSP370 | 151 (-27 / 349) | 38 (-0.7 / 91) | 15 (0.7 / 35) | 28 (-3.8 / 68) | 3.1 (-0.5 / 7.8) | 4.4 (-0.2 / 10) |
| | SSP585 | 168 (-18 / 373) | 45 (2.8 / 103) | 17 (1.4 / 38) | 34 (-0.8 / 79) | 3.8 (-0.2 / 9.1) | 5.5 (0.4 / 12) |
| M2 | SSP126 | 116 (-47 / 281) | 20 (-11 / 59) | 2.2 (-7.3 / 18) | 22 (-6.2 / 59) | 1.9 (-1.2 / 5.2) | 3.7 (-0.8 / 9.5) |
| | SSP245 | 119 (-45 / 287) | 23 (-9.3 / 64) | 9.4 (-4.6 / 32) | 25 (-4.4 / 64) | -0.25 (-2.3 / 2.2) | 4.3 (-0.4 / 10) |
| | SSP370 | 126 (-44 / 302) | 28 (-7.5 / 74) | 18 (0.9 / 44) | 29 (-2.6 / 70) | 1.04 (-1.6 / 4.0) | 5.0 (-0.2 / 12) |
| | SSP585 | 139 (-38 / 324) | 36 (-3.2 / 87) | 27 (8.8 / 54) | 37 (1.1 / 37) | 0.86 (-1.8 / 3.9) | 5.9 (0.3 / 13) |
| M3 | SSP126 | 81 (-69 / 238) | 3.1 (-22 / 34) | -4.04 (-10 / 2.6) | 14 (-12 / 49) | 0.0 (-2.4 / 3.2) | 2.5 (-1.4 / 7.8) |
| | SSP245 | 80 (-69 / 238) | 1.8 (-25 / 26) | -2.4 (-8.6 / 5.3) | 14 (-12 / 49) | -1.9 (-3.4 / -0.04) | 2.9 (-1.2 / 8.5) |
| | SSP370 | 76 (-74 / 239) | 1.4 (-25 / 28) | -1.9 (-8.3 / 5.9) | 17 (-11 / 53) | -1.7 (-3.4 / 0.26) | 4.1 (-0.5 / 10) |
| | SSP585 | 88 (-67 / 261) | 4.5 (-21 / 38) | 0.6 (-6.6 / 10) | 22 (-8.4 / 63) | -1.5 (-3.2 / 0.5) | 4.4 (-0.4 / 11) |
| M4 | SSP126 | 81 (-65 / 240) | 2.2 (-23 / 35) | -3.8 (-9.8 / 3.0) | 15 (-12 / 50) | -0.17 (-2.7 / 3.4) | 2.6 (-1.4 / 8.1) |
| | SSP245 | 80 (-66 / 240) | -3.2 (-25 / 26) | -2.5 (-8.9 / 5.5) | 15 (-11 / 50) | -2.0 (-3.5 / 0.2) | 3.0 (-1.1 / 8.7) |
| | SSP370 | 76 (-71 / 240) | -2.8 (-25 / 28) | -1.5 (-8.2 / 6.6) | 18 (-9.9 / 55) | -1.9 (-3.5 / 0.6) | 4.4 (-0.5 / 11) |
| | SSP585 | 89 (-63 / 262) | 3.7 (-21 / 39) | 0.99 (-6.5 / 11) | 23 (-7.4 / 65) | -1.7 (-3.3 / 0.95) | 4.6 (-0.3 / 11) |
| M5 | SSP126 | 91 (-45 / 219) | 3.4 (-23 / 33) | -3.5 (-9.8 / 2.9) | 17 (-8.2 / 47) | -0.35 (-3.5 / 3.0) | 2.9 (-0.8 / 7.4) |
| | SSP245 | 92 (-45 / 222) | 1.5 (-26 / 26) | -2.0 (-8.8 / 5.8) | 17 (-8.1 / 48) | -2.1 (-4.3 / -0.09) | 3.1 (-0.5 / 8.0) |
| | SSP370 | 90 (-48 / 225) | -1.0 (-25 / 27) | -1.2 -8.4 / 6.3) | 19 (-6.7 / 53) | -1.9 (-4.3 / 0.21) | 4.5 (0.2 / 9.9) |
| | SSP585 | 103 (-41 / 246) | 4.9 (-21 / 37) | 1.4 (-6.7 / 9.9) | 25 (-3.9 / 62) | -1.7 (4.1 / 0.6) | 4.7 (0.28 / 11) |

*Author contributions.* T. L. Smallman, D. T. Milodowski and M. Williams created the experimental design. TLS ran CARDAMOM and 595 analyses the outputs. G. Koren created the ensembles of Carbon Tracker atmospheric inversions. TLS led the writing with inputs from DTM and MW. All authors contributed to the writing of the paper.

*Competing interests.* The authors have no competing interests.

*Acknowledgements.* T. L Smallman, D.T. Milodowski, E. Soursa Neto and M. Williams were funded primarily by the UK Space Agency through the Forests2020 project (https://ecometrica.com/space/forests2020, last accessed 23/10/2020). T. L Smallman, and M. Williams 600 were additionally supported by the UK's National Centre for Earth Observation; M. Williams and D. T. Milodowski was additionally supported by the Newton Fund through the Met Office Climate Science for Service Partnership Brazil (CSSP Brazil). M Williams also





received funding from the Royal Society. This work has made use of the resources provided by the Edinburgh Compute and Data Facility (ECDF) (http://www.ecdf.ed.ac.uk/). LAI information was generated by the Global Land Service of Copernicus, the Earth Observation programme of the European Commission. The LAI product is based on SPOT-VEGETATION 1km data (copyright CNES and distribution

by VITO). G. Koren was funded by the ERC ASICA project (649087) and the inversions were carried out using a grant for computing time from NWO (SH-312-14). CarbonTracker Europe results provided by Wageningen University in collaboration with the ObsPack partners (http://www.carbontracker.eu). The authors thank Professor Wouter Peters for useful conversions on the use of atmospheric inversion analyses as an evaluation dataset.



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
