# Peer review of "Parameter uncertainty dominates C cycle forecast errors over most of Brazil for the 21st Century"

_Earth System Dynamics, 2021_

## Author Comment (AC1)

**Authors response to comments:**

Parameter uncertainty dominates C cycle forecast errors over most of Brazil for the 21st

Century

We thank you for your positive and constructive comments which we believe improve the manuscript. Below we deal with reviewer 1's comments in turn. Reviewer comments will be shown in red. Our responses to reviewer comments will be shown in normal text while new text intended for addition to the manuscript will be shown in *blue italics*.

**Reviewer 1:**

The study of Smallman et al. presents a model-data integration study where a suite of terrestrial ecosystem models of increasing complexity is inverted and evaluated using spatially resolved data across Brazil. The finding that already with quite simple models, parameter uncertainty is more important than model structural uncertainty and uncertainty in forcing data has a large impact of the earth system science and is worth to be published.

Thank you for your supportive comments.

I enjoyed reading the beginning of the paper and appreciated the well designed study using multi-model, multi-biome, site-specific spatially resolved setup, and varying input data for future climate scenarios, within a fully Bayesian inversion setting.
However, I got disappointed when I more closely inspected Table 2. Even with the simplest model, the confidence intervals of the predictions are so large, that only vague

and general statements or conclusions can be drawn from the results. All the elaborations

and conclusions on model complexity and structural error would be a really good

presentations, if the results were more constrained. However, with the large uncertainties

I would recommend to only summarize them and omit the detailed presentation, because

they are base on vague ground.

Thank you for this comment. Uncertainties estimated at pixel level are directly derived from the

pixel specific ensemble of accepted parameters. These uncertainties are presented in maps

indicating where our analyses are consistent with independent evaluation datasets (e.g. Fig. 4)

and spatial differences in structural vs parametric uncertainty (Fig. 7, S15).

As you note our uncertainties at both national and pixel level appear surprisingly large on first

viewing. However, we argue that in reality it is the uncertainty estimates of other individual

estimates (e.g. FLUXCOM and CTE) which are likely underestimating. For example, as we

noted in the introduction, the range of global GPP estimates by independent methods varies

between 80-170 PgC yr-1 (Shao et al., 2013; Joiner et al., 2018; Jung et al., 2020). If we

assume a median estimate of 130 PgC that suggests an uncertainty of ~69% in just one

component of the carbon cycle, likely the best known. Similarly, the range of values in the global

land sink from independent atmospheric inversion frameworks vary on the order of 1 PgC yr-1

similarly representing an uncertainty of 50-60 % relative to our best estimates of the mean

terrestrial C exchange (Friedlingstein et al., 2020, https://doi.org/10.5194/essd-12-3269-2020).

CARDAMOM is simulating the whole terrestrial biogenic C-cycle, not just a component. Despite

this greater challenge CARDAMOMs national scale estimate of parametric uncertainty for GPP

is 51-77 % (Table 2) - similar to that found between independent estimates. In this context we

do not think it is surprising that our estimate of uncertainty for net exchange is large or even larger than the mean magnitude. Moreover, we argue that as other studies provide their ecological interpretation - in the face of unknown uncertainties - it is appropriate that we should also provide interpretation. However, we understand that this interpretation must be clearly caveated.

Uncertainties associated with Brazil-wide totals (Table 2, Fig. 5, 6) were estimated assuming errors are fully correlated. We consider it to be the most conservative approach given our lack of robust information on the spatial correlations of uncertainties. We should have noted this in the text. The corresponding alternate extreme assumption, to assume all pixel errors are fully uncorrelated, yields unrealistically small uncertainties. The truth will lie somewhere between these two approaches. For example, M1 GPP shown in Table 2 (fully correlated uncertainty propagation) is 17.7 (9.8 / 23.4) with the uncorrelated assumption being 17.3 (17.0 / 17.6) TgC yr-1. Note the small difference in the mean is due to pixel-level uncertainties not being Gaussian and undersampling the possible between-pixel ensemble combinations. Were we to present the uncorrelated uncertainties this would imply that parametric uncertainty is smaller than model structure which is inconsistent with the pixel level information (Fig. 7, S13), supporting our choice of presenting the fully correlated assumption.

To address you comments we propose to:

1) Provide improved context on the relative uncertainties represented by the range of independent estimates found in the literature (e.g. GPP and Net C exchange). Minor changes throughout the introduction and discussion will be made to clarify this.

2) To add the following paragraph to the CARDAMOM description (Section 2.1) to clarify our uncertainty propagation assumptions. "*Pixel-level uncertainties are estimated directly from the CARDAMOM retrieved ensembles of parameters, and their model generated C stocks and fluxes. However, we lack a robust understanding of how uncertainties are correlated in space, making the propagation of uncertainties from pixel-level to Brazil-wide challenging. Assuming an intermediate value would lead to an arbitrary estimate of uncertainty while assuming either fully-correlated or -uncorrelated uncertainties leads to either an over- or under-estimate in Brazil-wide uncertainties respectively. To be conservative, here, we assume uncertainties are fully-correlated when propagating from pixel to Brazil-wide estimates. To allow for non-Gaussian distributions in the pixel-level ensembles we assume that the fully correlated assumption is approximated by aggregating the pixel-level 5 % and 95 % quantiles across Brazil as previously done (e.g., Exbrayat et al., 2018b). Again to be conservative we will only discuss in detail between-model differences which are also supported in the pixel-level estimates.*"

3) Figure legends showing time series aggregates will include the following text. "*Note that uncertainties were propagated from pixel level to Brazil-wide totals assuming fully-correlated uncertainties.*"

4) Reduce the detail given when discussing the model structural impact on Brazil wide totals unless these are also substantial by the spatial patterning.

5) Add additional text in the manuscript that highlights the consistency / robustness of our conclusion from both the national scale estimates and at pixel-level which are not impacted by the poor knowledge of how to propagate uncertainty between resolutions.

Instead of studying increasing model complexity, the results show that the data is already not enough to constrain the simplest model variant. Statements like LL259 "simulated NEE was consistent with CTE ensemble at the 90% CI" does not tell me much about the goodness of the model, if the CI range is 400% of the median estimate. In order to defend the insights despite the large uncertainty, the posterior density of the parameters in comparison to the priors should be provided as a supplement or appendix.

Thank you for this comment. The issue of signal to noise is an important one which we have not directly dealt with in this paper as we consider it to be out of scope. However, it is important to consider the relative uncertainties of our model analysis versus the observational information available. For example, proportionally the CTE ensemble has a mean uncertainty of 3000 % due to much of their analysis being near zero while the DALEC models vary between 1200-3100 %, which we accept is currently biased towards a net uptake of C. However, the mean absolute uncertainty value for NEE from the CTE ensemble (0.5 gC m-2 day-1) is less than the DALEC models' (3.1-3.5 gC/m2/day).

We now provide in the appendix maps (figure below) for each model showing 1-posterior:prior ratio (i.e. uncertainty reduction in the posterior relative to the prior) for the process parameters, initial conditions (C and where appropriate water) and parameters which are common to all models. A new table will be added to the appendix containing the Brazil wide mean posterior reductions per model for all parameters.

[Figure]

New Figure - shows the mean proportional reduction in the posterior parameter ranges relative to the uniform prior ranges sampled from by CARDAMOM. Columns show the results for each model M1-5. The top row shows the posterior reductions for the process parameters, the middle row shows the initial conditions (i.e. C and H2O pools at t=1) and the bottom row shows the parameters which are common across all models. A complete list of common parameters is provided in a new supplementary table.

In the main manuscript Sect. 3.1 Calibration Constraints, we will add the following text.

"*The reduction of parameter uncertainty between the 90 % confidence interval and the prior range is highly variable across Brazil, between parameters and to a lesser extent models (Table A2,3, Figure A7). The reduction in the parameter posteriors relative to the prior bounds (1-posteriorCI90:prior range) varies between model (M2 = 0.55, M5 = 0.46; Table A2) but with much larger variability between parameters (Rhet coefficient = 0.12, initial soil = 0.96) and*

*across Brazil (Caatinga = 0.62-0.7, Amazon = 0.42-0.5; Figure A7). The spatial pattern across Brazil broadly follows the spatial distribution of precipitation (Figure A2). The greatest reduction in posterior parameter uncertainty is typically achieved in M2 with the lowest in M5 and broadly similar values in M1, 3, 4. Parameters related to initial C conditions and canopy phenology are best constrained, as expected given the majority of observations directly relate to these parameter groups, while NPP allocation and turnover / decomposition related parameters are least constrained in the posterior (Table A3)."*

The main conclusion about the dominance of parameter uncertainty is strengthened by this large uncertainty and should be published, with a much shortened presentation of the (to my opinion vague) comparison across model structures.

We appreciate the concerns of over-interpreting our results. But we do think it is appropriate to provide some interpretation around model structures to identify how model complexity affects analyses. Such interpretation would be expected in a paper using a classic land surface model approach with predefined parameters. However, in those circumstances model parametric uncertainty would be unknown. However, we will revise the results section to focus on cases with robust differences. To provide greater context, we highlight the differences between our analysis and information gained from a traditional land surface model analysis. We will add the following sentence to paragraph 3 of the introduction.

*"However, as TEMs typically lack information on their parametric uncertainty it remains unclear whether model differences are driven by different parameter estimates or model structure."*

An alternative route, which requires a larger reanalysis effort, is based on the claim of the authors that the model can be constrained by repeated EO observations of biomass. In addition to the current model inversion, I suggest generating an artificial observation of this biomass data stream using the most complex model variant add noise and some slowly-changing bias and repeat the inversion including this artificial data. If the uncertainties decrease as much, the presentation about model structure could be kept, but based on this new (artificially) more constrained inversion results.

This is a very good idea and one we have discussed previously. The impact of assimilating repeat biomass observations has been quantified at site scale in a previous study (Smallman et al., 2017). However, we consider the synthetic study of repeat biomass estimates to be out of scope for the current study. Our primary focus here has been to quantify the relative contributions of uncertainty. This focus required a novel approach to assessing land surface models using ensemble based approaches which are typically not used for large scale land surface models due to their computational complexity. We will be focusing on testing the information content of repeat biomass maps explicitly in a subsequent study.

Specific comments

To gain an conception about the computational effort: At how many pixels was the model Inverted?

Very good question - this should have been included in the manuscript. There are 702 pixels. This information will be added to the opening sentence of the methods section (Sect. 3).

Line 220: It did understand how "future climate is imposed by determining the anomaly from the end of the analysis until 2100". Please, extend this explanation.

We apologise that we have not provided sufficient detail on how we created the future climate drivers for our analysis. L220 will be expanded to the following.

"*The contemporary meteorology from observations differs from that generated in the climate models used to project future climate. As a result there are step changes in drivers between historical and future climate, impacting the simulation of the carbon cycle in an unrealistic manner. To avoid these step-change impacts future meteorology is imposed as an anomaly relative to 2018. Specifically, each month of the future meteorology extracted from the UKESM has the corresponding month from 2018 subtracted creating the anomaly time series, i.e. each month of 2018 anomaly would be equal to 0. The anomalies are then added to the absolute values of the monthly values from 2018 from the calibration meteorology time series but with sanity checks to prevent negative values in positive definite variables.*"

Line 223: I assume the model structural uncertainty was estimated for each climate scenario separately (and the climate uncertainty for each model variant separately), right? Or does the "between model range" span across all climate scenarios?

Sorry we didn't clarify this point. We estimated the model structural and parameter uncertainty separately for each climate scenario and averaged across climate change scenarios to account for potential differences in model structural or parametric response under different climates. The text will be modified to clarify this situation.

*"Both parametric and structural uncertainties were estimated for each climate change scenario and then averaged across scenarios to provide an overall estimate"*

Fig 4: In my opinion, the stippling (indicating consistency within confidence range) does not tell much when considering the large uncertainties.

We believe that it is important and informative to indicate where across Brazil our model ensemble is consistent with observations. The stippling shows us that even with the large uncertainties some parts of the ensemble include a consistent assessment of carbon dynamics when considering the projections into the future. To provide appropriate context we highlight that our uncertainties are large and show where and by how much our analyses are biased (Figure A8) with respect to the independent data rather than considering a single metric of evaluation.

We will rebalance our results to provide a more comprehensive evaluation to avoid an over reliance on the stippling metric. We will strengthen our definition of consistency. In the existing framework we assumed consistency based on the mean flux over the relevant time period for each independent dataset. We will use a stricter measure which requires > 90 % of time steps in the overlapping period between the model simulations and independent estimates. This will provide a more granular interpretation of consistency. We realise also that our definition of consistency is not clearly defined in the text. We will add the following text in Section 2.4 to correct this omission.

*"A key evaluation metric is the degree of consistency at pixel level between the DALEC models and the independent historical evaluation data. We define consistency as the pixel-level*

*ensemble of DALEC C-cycle estimates overlapping independent observations at >90 % of observed time steps*"

Fig 5: putting all the labels the center panel confused me first, I suggest putting the observation legends to the panels. Almost all the streams are encoded by color, which for me were difficult to read.

It is very important to us that our figures are as easy to read as possible. We would like to avoid adding additional complexity to the plots by adding figure specific legends or boxing the legend. We have moved the legend to the top panel to make it more noticeable and changed the line type for each of the models to help add greater distinction rather than colour alone.

L286, 294: Why is the NEE not improved with model complexity, if fire is improved and makes up 3 to 30% of NEE?

This is a very good question, the answer to which we have not made clear in the manuscript. NEE is not improved because of compensating changes in autotrophic and heterotrophic respiration (Table 2). Between M5 and M4 fire emissions increase, thus reducing the bias with independent estimates. However, at the same time respiration decreases providing a compensating impact on net exchange. This behaviour reinforces the need for greater constraint in our analysis using a range of new data such as those we summarise in the discussion section. To highlight the compensation challenge the following text will be added to the manuscript after L294.

*"Despite the improvement in estimation of C emissions due to fire there is no corresponding improvement in NEE or NBE due to compensating changes in both autotrophic and plant respiration (Table 2). This result highlights the need for greater overall constraint on the C-cycle, for instance independent estimates of respiratory fluxes."*

L315: How did you assign priors to the MRT parameter? The sentence suggests a Normal distribution that includes also negative values. A lognormal prior would be more appropriate and you could report the multiplicative moments of the posterior and avoid negative residence times.

We do not provide prior estimates (with Gaussian uncertainty) for any C pool MRT parameters. Instead, all MRTs are provided only with a uniform prior range of ecologically plausible values from which the parameter proposals are drawn. Therefore, there are no negative values being proposed for MRT. In section 2.1 the following sentence has been revised to clarify

*"Each chain assesses 100 million parameter proposals, drawn from uniform prior ranges, from which a sub-sample of 1000 accepted parameter vectors are stored."*

Sec 3.3. reads lengthy. Are all the details necessary in the main text. I have, though, no specific suggestion how to shorten.

We will work to simplify this text to in a manner consistent with our other responses, e.g. ensuring that we highlight relevant results that are robust and support the interpretation.

L331: Hints to model error. Thanks for the discussion at L462ff, that could be referenced

at this point. For me it did not become clear, how biomass removal was accounted for in the DALEC simulations, and the future scenarios.

We will clarify how biomass removal is imposed in DALEC in the model description. We will also add into the SI a figure (see response to reviewer 2 for figure) which shows how each of: biomass removal, fire and "natural" /  unexplained turnover contributes to the overall estimated MRT. Critically we will see how these differing components vary across Brazil..

L 347: Can you quantify "most likely"? Can you infer p(deltaBiomass(t) > 0) from the posterior?

Yes we can. Our model ensemble estimates that in SSP 2-4.5W/m2 Brazil's total biomass has a likelihood of 73-85 % of increasing while the likelihood of total DOM increasing is 64-84 %. We will add a complete table of this information for all scenarios to the appendix and the following line in paragraph 2 of Section 3.4.1.

*"Using our ensemble based approach we estimate that the likelihood of a net increase of C in biomass by 2100 is 73-85 % while the likelihood of a net accumulation in DOM is 64-84 % (Table A6)."*

L 450: may replace "a function of three factors" by "There are three possible interacting Explanations"

Done

---

## Author Comment (AC2)

**Authors response to comments:**

Parameter uncertainty dominates C cycle forecast errors over most of Brazil for the 21st
Century

We thank you for your positive and constructive comments which we believe improve the
manuscript. Below we deal with reviewer 2's comments in turn. Reviewer comments will be
shown in red. Our responses to reviewer comments will be shown in normal text while new text
intended for addition to the manuscript will be shown in *blue italics*.

**Reviewer 2:**

The manuscript " Parameter uncertainty dominates C cycle forecast errors over most of
Brazil for the 21st Century" by Smallman et al. presents projections of Brazilian carbon
cycling using a model-data hybrid approach. A terrestrial ecosystem model with varying
degrees of complexity is constrained by remote sensing data before being used to quantify
the evolution of carbon stocks into the future. Overall, I think the methodology is robust
and I commend the approach, in particular separating and quantifying parameter,
structural, and climate uncertainty on future stocks. The paper is well written, flows well,
and addresses an important topic with substantial and robust conclusions.
Therefore, I would recommend the manuscript to be published after the following
comments have been addressed:

Thank you for your positive comments.

In general, the paper has detailed reporting of results, however the explanations of

model behaviour are sometimes incomplete. This lets down the previous good work

that comes beforehand. For example:

L406 - Can you explain why RT increased?

Thank you for your question. Our discussion tries to balance giving as full an investigation of the

model outputs as we can without over-interpreting given the large uncertainties associated with

our analyses. However, in this instance we didn't fully explain the likely pathway of the response

as it may be specific to our modelling framework. The reduction in GPP (M3-5) coincides with

the inclusion of the water cycle, resulting in water availability limits. At the same time

CARDAMOM still aims to match the available observational information within the ecologically

consistent parameter space defined by the priors and ecological and dynamical constraints.

Thus, with reduced C inputs CARDAMOM responded by selecting parameters with longer

residence times (i.e. reducing outputs) to maintain the C-balance consistent with the available

observations.

L407 - I find it interesting that here the net flux correlates with GPP, but the long term

net changes have low correlation with GPP. Can you explain?

Good question. GPP is a major component of C exchange particularly on seasonal timescales.

However, on longer time scales the trajectory is determined by the residence times of slow

turnover pools, i.e. wood and soil. This effect has been found in previous modelling studies (see

our introduction). What we have done here demonstrates both responses within an

observationally constrained framework. This result also shows our framework should also be

able to address this knowledge gap if we can assimilate high-confidence repeated AGB estimates.

We don't explicitly highlight the different processes dominating on differing time scales but we agree that it would be a benefit to do so. We will add the following text to make that connection when discussing the correlation between biomass change and residence time in the discussion (Section 4.3, paragraph 1)

*"The analysis of long term C trajectories contrasts with the correlation between GPP and net C exchange during the calibration period. This contrast highlights the importance of considering the timescale of change that is of interest, with wood and soil residence times driving long term net C exchanges."*

L414 - Can you explain why Amazon flux is overestimated?

Despite the quasi-steady state assumption within the Ecological and dynamical constraints (EDCs) we find that CARDAMOM analyses tend to follow the trend in LAI, particularly in cases where there isn't a further indication of the C-balance e.g. high levels of disturbance. Across much of Brazil LAI observations are increasing over the calibration period. Finally, forest losses used to force the model (Global Forest Watch) are likely underestimated due to lack of degradation observations, which are significant (Milodowski et al., (2017) doi: 10.1088/1748-9326/aa7e1e). In the discussion we argue that the key missing information is repeated AGB estimates or other information which imposes a bulk C-balance constraint.

It would help the reader if more insight was offered into what drives model responses.

The introduction stresses the importance of land-use change and fire for the Brazilian carbon cycle, but there is little discussion on how these disturbances will influence future carbon stocks. Are they important? If not, why not? If they are, then how important?

You are quite right, we do stress the importance of disturbance as a driver of the C-cycle and impacting the community compositions of ecosystem traits. Much of the modelled response to future climate change is divided between the Amazon / Atlantic forests or dry regions, while disturbance (as observed in our current drivers) is focused on the arc of deforestation. We will add a new figure to the SI which explicitly partitions across Brazil the contribution of fire and biomass removals on the woody mean residence time. The following text will also be added to the results section to clarify the contribution of disturbance across the contemporary C-cycle.

*Moreover, our analysis allows us to partition MRT into its constituent drivers, i.e. natural, fire and biomass removal, which indicates that given currently available drivers disturbance is only a major determinant of MRT across the Amazon Cerrado boundary (Figure A14).*

[Figure]

New Figure - shows the fractional contribution of fire (driven by MODIS burned area), biomass removal (driven by global forest watch) and the remainder (i.e. natural / unexplained).

Finally, we will clarify how we extracted fire and biomass removal drivers from the future climate change scenarios with the following text in Section 2.5.

"Disturbance due to forest harvest is driven by the management scenarios associated with each SSP. However as DALEC does not represent land cover types we neglect land use change in the drivers. Thus any forest which undergoes biomass removal subsequently remains a forest and is allowed to regrow. Finally, as we currently lack a predictive model of fire in DALEC (i.e. we drive fire with EO burned area), we extended observed fire for the contemporary period into the future simulations."

Also, it is not clear if/how land-use change is implemented in your future simulations. Is this implicit in the mortality parameter? It would be good to clarify this.

As indicated above we explicitly partition between natural, fire and biomass removal drivers of residence time. As described above we will add text clarifying both in the model and climate change source description how we use these drivers.

The writing style can be improved in places. There are a few dense paragraphs (that can sometimes read like a list), e.g. in Sections 3.3 and 3.4.3, there is often a detailed comparison between the five models that doesn't add much value. Can the values be moved to a table and then the text just discusses the most important parts? E.g. "Carbon allocation to wood as well as wood turnover rates determine future biomass stocks... Relative contribution of allocation and turnover varies across biomes…".

We will work to refine our presentation, divide paragraphs and remove overly descriptive text to improve the flow of the manuscript and focus on aspects picked up on in the discussion.

Also, at times the paper sounds quite negative. E.g. in Section 3.4, I feel some value is lost when all your results are caveated with "uncertainty is larger than predicted changes" or (L299-305) - "...insufficient observational constraint to confidently determine the sign of NBE or soil C dynamics. The same was largely true for wood stock dynamics..". I appreciate the uncertainty is important (and the focus of the paper), and certainly needs to be addressed, but this large uncertainty seems to undermine any projections you make about future stocks.

Thank you for this comment. We aim to be clear and open about the size of our uncertainties and their implications. However, we also recognise that we have not adequately set the context against other studies which discuss model structural differences but ignore their parametric uncertainty. We will seek to rebalance the tone of the manuscript as well as adding additional context on the size of uncertainties in the associated datasets. See previous comments for the first reviewer for examples.

Can M1-M5 capture the full extent of model structural uncertainty? You start to discuss this (L467) but in my opinion, I don't think this is a strong enough justification. Further, you state "...limited sensitivity of C cycling to future soil moisture stress.", which is at odds with drought experiments that suggest increased mortality and reduced productivity (e.g. ACL da Costa et al., 2010 - New Phytologist).

Thank you for your comment. We do not intend to fully capture the range of model structural uncertainty. Rather we wish to contrast parametric uncertainty relative to the impact of structural changes which are of a scale typically carried out in the TEM community and address specific

hypotheses. We have added text to clarify this in the introduction and in the text surrounding L467. Thank you also for highlighting that our analysis contrasts the results of the rainfall exclusion experiment. We have included this reference in our analysis. The paragraph starting L467 now read following:

*Variations in C storage linked to model structure were smaller than those linked to model parameterisation, except in specific areas of Brazil (Caatinga; Figure 7, S15). The selection of five model structures was limited by our choice, so it is perhaps not surprising that the parameter calibration, which allows for multidimensional variation over broad priors, contributes more variation to projections than does model structural variability. However, the variation in model structure was designed to test whether hypothesised key processes were important in projections and similar to the kinds of developments which are tested and interpreted in ESMs and / or TEMs. For instance, we used models with and without a water cycle simulation to test the importance of carbon-water feedbacks in projections of C storage to 2100. Models M3-5 included dynamic simulation of soil moisture changes and its interactions with canopy processes. Projections with these models thus included the potential development of soil moisture stress, with an impact on GPP. Models M1 and M2 had no direct effect of soil moisture on C cycling. This soil moisture feedback on GPP only manifested itself in projections for north east Brazil, the driest part of the country, in the Caatinga biome, and some nearby parts of Cerrado (Figures 4, A8). This feedback does have an impact on projected C storage (Figure 7; Table A5), but these effects are of similar or less magnitude to parameter uncertainty. We conclude that for much of Brazil, outside of Caatinga, our model-data fusion shows a limited sensitivity of C cycling to future soil moisture stress. However, our modelled analysis contrasts the finding of the Caxiuana rainfall exclusion experiment which found drought enhanced tree mortality and reduced productivity (da Costa et al., 2010). Our result is likely a result of $CO_2$*

*fertilisation leading to reductions in plant water demand that are explicit in both ACM GPP*

*models. However, it is possible that land surface models like DALEC are overestimating CO2*

*fertilisation effect Wang et al., 2020) and/or by using time invariant parameters (i.e. traits) we*

*are neglecting the impact of species change (i.e. biodiversity shifts) on ecosystem response.*

*Collectively, these results highlight the need for further evaluation and refinement.*

Minor comments:

- Looking at Figure A7, M5 looks like it still has a large fire bias similar to the other

models, so can Can M5 say anything about the role of fire on future C stocks in the future?

You are correct that the spatial biases remain substantial in M5. What our analysis can tell us is

that wood litter is an important combustible C pool which allows us at national scale to improve

our C emissions. However, we still need more information to help constrain wood litter and other

combustible pools. We argue that high confidence repeated AGB estimates will play a vital role

in constraining both live biomass and dead organic matter dynamics (see Smallman et al., 2017

for site level experiment). More information on litter pools would be valuable too.

- FLUXCOM-RS+METEO is known to underestimate IAV. One option would be to use

FLUXCOM-RS which uses interannual satellite LAI as an input and generally shows large

IAV and trends. This could improve the comparison in Figure 5.

You are quite right to point out that our independent datasets will have limitations and bias' of

their own. To improve the GPP contrast we have added two further independent GPP products

which use distinct approaches to that used in FLUXCOM and thus giving an improved representation of observational uncertainty associated with GPP. The newly included estimates are Copernicus GPP product (i.e. the same data provider as LAI assimilated here) and FluxSatv2 (Joiner and Yoshida. 2021, doi: https://doi.org/10.3334/ORNLDAAC/1835). Figure 5 and the GPP consistency metrics have been updated to account for this change.

L46-47 - Missing words?

We recognise that this sentence can be improved. We will revise this to:

"*However, estimated responses to environmental change are sometimes contradictory between studies indicating model ensemble (i.e. model) specific conclusions*"

L299 - "Carbon dynamics of wood, not soil, is the primary driver of net exchange." - Does this depend on timescales? Over the course of a century, won't slower (but potentially larger) soil C dynamics play a bigger role?

Yes time scales do make a difference. While wood as a relatively slow pool makes a major contribution to C balance. The reported values are assessing the contemporary period (2001-2017). Soil stock changes are highlighted in the future scenario element but we do not generate the corresponding correlation. Furthermore, as highlighted in the future simulations we show that both live and dead organic matter change substantially over longer time scales and we quantify the dominant controlling traits.

L394 - Missing words?

We apologise for that sentence. We have now clarified this and related sentences.

Increasing model complexity by including a wood litter pool (M5) reduces bias (by ~65 \%) between total fire emissions estimated by DALEC and independent estimates (Table 2, Figure 5). However, the spatial consistency remains largely unchanged with a persistent spatial disagreement across the southern Cerrado / Amazon boundary (Figure 4) and the DALEC estimated Brazil-wide fire emissions uncertainty increased on the addition of a wood litter pool (Table 2).

L395 - Increased from what? Due to increasing model complexity? Over time?

We apologise for this sentence we have revised it to:

"*However, the DALEC estimated Brazil-wide fire emissions uncertainty increased on the addition of a wood litter pool (Table 2) and there remains a persistent spatial disagreement across the southern Cerrado / Amazon boundary (Figure 4).*"

L394-402 - It would be good to place wood litter in the context of other drivers of fire. Is wood litter the most important variable if we want to correctly simulate fire? How about the local climate? Ignition sources?

You are correct that local climate and ignition source are important components of fire instance and emissions rather than just fuel load, as considered here. These topics are out of scope for the current study as we do not simulate the explicit impact of local climate (i.e. how dry is the

fuel load) or ignition. Our analyses are driven by satellite observations of burned area that integrate local climate and ignition source factors.

We will flag these issues in the discussion as you point out as components which are not addressed by this study.

L407 - Do you mean "GPP" instead of "net carbon uptake"?

No. A point we make on L410 is that NEE is becoming more negative (i.e. the net uptake is increasing) and that this is driven primarily by the positive trend simulated for GPP.

L454 - Why is it important that NPPWood is the second most important determinant?

Our results show that it's not all one or another. There is substantial co-dominance in many locations highlighting the need to target constraints on multiple components. However, we realise that this sentence is poorly written and will be revised in the following way

*Similarly, it is important to note that NPPwood is the second most important determinant of future dynamics of both biomass and DOM, and in many areas is co-dominant with MRTwood (Figure 8). Efforts to constrain estimates of both MRTwood and NPP allocation are thus critical for more robust predictions of C storage*

L456 - Don't some transient biomass maps exist already? e.g. ESA, or VOD based (Liu et al. 2015). L533 - But don't we have transient AGB and atmospheric NEE available now?

This is a good question and one that we considered out of scope to specifically pick up on limitations of current maps as approaches which address these shortcoming are already being developed. Briefly, the (ESA) CCI Biomass maps do exist for three years (2010, 2017 and 2018). However, the creators of these maps currently advise against using them for change detection due to their creation using different underlying field data and satellite information potentially introducing differing biases between years. We will note this in the discussion. The Liu et al., (2015) dataset derived using VOD remains an area to watch for us as the correct interpretation between plant water content and its biomass is yet to be fully refined. Finally, on the NEE / NBE side we choose to keep these data as independent evaluation to help provide a benchmark for future improvement. We will modify the beginning of the paragraph at L494 to highlight this choice but that assimilation is also a valuable opportunity.

---

## Author Response (AR2)

Authors response to editors minor corrections

Thank you for your comments and minor corrections for the manuscript. These will improve the paper and clarify several lines / figures

1. Figure caption for Fig. 3: please number the panels, e.g. a-d, and refer to them in your figure caption to which panels the description refers, i.e. the grey line (a-c) refers to the 1:1 line. Also, you need to explain the colours of the LAI time series where you compare them against the Copernicus data.

Done

2. Line 297: please remove "[]" at the beginning of the sentence.

Removed

3. Figure 8,9 is very small and its pattern hard to read. Consider increasing its size.

We have increased these in size

4. Line 455: do you mean "southern Amazon-Cerrado border"?

Yes, we have changes the sentence to your phrasing for clarity

5. Line 535: please revise this sentence or consider cutting in two: "While EO derived AGB maps do exist of different time periods (e.g. ESA CCI santoro et al., 2021) these estimates are typically created using different processing chains, calibration data, satellite etc. resulting in these estimates having different bias and error structures making them more challenging dataset to use at this time."

Yes, this sentence is a bit of a mouthful. We have divided this into two.